# Temporal proteomic analysis of HIV infection reveals remodelling of the host phosphoproteome by lentiviral Vif variants

Edward JD Greenwood*†, Nicholas J Matheson*†, Kim Wals, Dick JH van den Boomen, Robin Antrobus, James C Williamson, Paul J Lehner*

Cambridge Institute for Medical Research, Department of Medicine, University of Cambridge, Cambridge, United Kingdom

**Abstract** Viruses manipulate host factors to enhance their replication and evade cellular restriction. We used multiplex tandem mass tag (TMT)-based whole cell proteomics to perform a comprehensive time course analysis of >6500 viral and cellular proteins during HIV infection. To enable specific functional predictions, we categorized cellular proteins regulated by HIV according to their patterns of temporal expression. We focussed on proteins depleted with similar kinetics to APOBEC3C, and found the viral accessory protein Vif to be necessary and sufficient for CUL5-dependent proteasomal degradation of all members of the B56 family of regulatory subunits of the key cellular phosphatase PP2A (PPP2R5A-E). Quantitative phosphoproteomic analysis of HIV-infected cells confirmed Vif-dependent hyperphosphorylation of >200 cellular proteins, particularly substrates of the aurora kinases. The ability of Vif to target PPP2R5 subunits is found in primate and non-primate lentiviral lineages, and remodeling of the cellular phosphoproteome is therefore a second ancient and conserved Vif function.

*For correspondence: ejdg2@ cam.ac.uk (EJDG); njm25@cam.ac. uk (NJM); pjl30@cam.ac.uk (PJL)

†These authors contributed equally to this work

Competing interests: The authors declare that no competing interests exist.

Reviewing editor: Éric A. Cohen, IRCM-UdeM Chair of Excellence in HIV Research, Canada

## Introduction

Viruses hijack host proteins and processes to optimize the cellular environment for viral replication and/or persistence. Manipulation by viruses signposts critical pathways in viral pathogenesis and cell biology, and evolutionary pressure has led to conflict between cellular restriction factors (limiting viral replication) and viral countermeasures (overcoming restriction in vivo). We previously used multiplex whole cell proteomic analysis of Human Cytomegalovirus (HCMV)-infected fibroblasts to define expression time courses of viral and cellular proteins and identify novel proteins involved in the host-HCMV interaction, a technique we termed Quantitative Temporal Viromics (QTV) (*Weekes et al., 2014*). Here, we provide a comprehensive temporal proteomic analysis of HIV infection.

The HIV-1 'accessory proteins' Vif, Vpr, Nef and Vpu share a common ability to target cellular proteins for degradation (*Simon et al., 2015*; *Sugden et al., 2016*). Whilst dispensable for viral replication in vitro, they are essential for pathogenesis in vivo. Nef and Vpu are multifunctional adaptors which co-opt endolysosomal and proteasomal machinery to downregulate numerous plasma membrane proteins, including their canonical substrates CD4, tetherin and MHC class I. In contrast, although Vif and Vpr are known to target cytoplasmic and nuclear proteins for proteasomal degradation, relatively few cellular substrates have been reported.

The only known Vif targets are members of the APOBEC family of cytosine deaminases, which are otherwise incorporated into viral particles and act as dominant restriction factors causing hyper-

**eLife digest** About 100 years since it was first transmitted to humans, the Human Immunodeficiency Virus (HIV) infects almost 40 million people worldwide and causes more than a million AIDS-related deaths every year. It is therefore critical to understand how HIV has been able to multiply and spread, and why infection with HIV causes AIDS. In the evolutionary "arms race" between viruses like HIV and the cells they infect, viruses try to enhance their ability to multiply, and cells try to resist. These interactions emphasise the processes that are most important for cells and viruses, and suggest new ways to treat HIV and other viral infections.

Proteomics is the large-scale study of molecules known as proteins, the critical building blocks of both cells and viruses. Greenwood, Matheson et al. use proteomics to measure how the abundance of proteins in human cells change during HIV infection, and identify new interactions between the virus and its host. The experiments distinguish more than 6500 proteins, and reveal that an HIV protein called Vif destroys several key components of a cellular protein called PP2A. Previous studies have demonstrated that PP2A plays a critical role in regulating the activities of numerous other proteins and processes in cells. Greenwood, Matheson et al. further show that other HIV-related viruses that infect monkeys, apes and even sheep can also counteract PP2A, suggesting that this interaction has been important during host and virus evolution.

The next steps following on from this work are to find out why HIV attacks PP2A, and whether drugs that interfere with this interaction may help to treat HIV infection. A future challenge will be to investigate how HIV interacts with other cellular proteins highlighted by the proteomics approach.

mutation of the HIV genome (**Desimmie et al., 2014**; **Malim, 2009**). Whilst Nef, Vpr and Vpu are found exclusively in primate lentiviruses, Vif is found in four of the five extant lentiviral lineages, infecting primate, feline, bovine and small ruminant hosts (**Gifford, 2012**), and the ability to target cognate host APOBEC proteins is conserved across Vif variants from all these diverse lineages (**Larue et al., 2010**).

Cellular proteins regulated by HIV have generally been identified using non-systematic, candidate approaches. We recently used a different, unbiased plasma membrane proteomic approach to reveal >100 previously unsuspected cell surface proteins depleted by HIV-1, including novel Nef (SERINC3/5) and Vpu (SNAT1) targets (**Matheson et al., 2015**). Whole cell proteomic studies of HIV-infected cells have been variably hampered by limited proteome coverage, asynchronous infections and confounding by the presence of bystander (uninfected) cells (**Supplementary file 1**). Consequently, it has been difficult to attribute changes in protein levels to expression of specific viral genes, and intracellular proteins targeted by HIV accessory proteins have not been discovered in this fashion.

In this study, we extend our tandem mass tag (TMT)-based temporal proteomic approach to describe global changes in HIV-infected T cells, comprising expression time courses of >6500 proteins. We cluster proteins according to their patterns of temporal expression, and identify >100 cellular proteins regulated by HIV, including candidate resistance/restriction factors and HIV accessory protein targets. To test the utility of our approach, we focus on proteins depleted with similar kinetics to APOBEC3C, and confirm the B56 family of regulatory subunits of the key cellular phosphatase PP2A (PPP2R5A-E) to be novel Vif targets. We use large-scale quantitative phosphoproteomics to demonstrate Vif-dependent remodelling of the cellular phosphoproteome during HIV infection, and show that, along with APOBEC proteins, antagonism of PP2A-B56 is an ancient and conserved Vif function.

## Results

### Systematic time course analysis of protein dynamics during HIV infection

To gain a comprehensive, unbiased overview of viral and cellular protein dynamics during HIV infection, we analysed total proteomes of CEM-T4 T cells infected with HIV. As previously described

(*Matheson et al., 2015*), cells were spinoculated with Env-deficient, VSVg-pseudotyped virus at an MOI sufficient to achieve a synchronous single round infection with <10% uninfected bystander cells. We exploited 6-plex TMT labelling to quantitate 6538 proteins in whole cell lysates of uninfected cells (0 hr), at four timepoints following HIV-1 infection (6, 24, 48, and 72 hr), and in cells infected for 72 hr in the presence of reverse transcriptase inhibitors (RTi) (*Figure 1A*). The complete dataset has been deposited to the ProteomeXchange consortium with the dataset identifier PXD004187 (accessible at http://proteomecentral.proteomexchange.org) and is summarised in an interactive spreadsheet (*Figure 1—source data 1*), which allows generation of temporal profiles for any quantitated genes of interest.

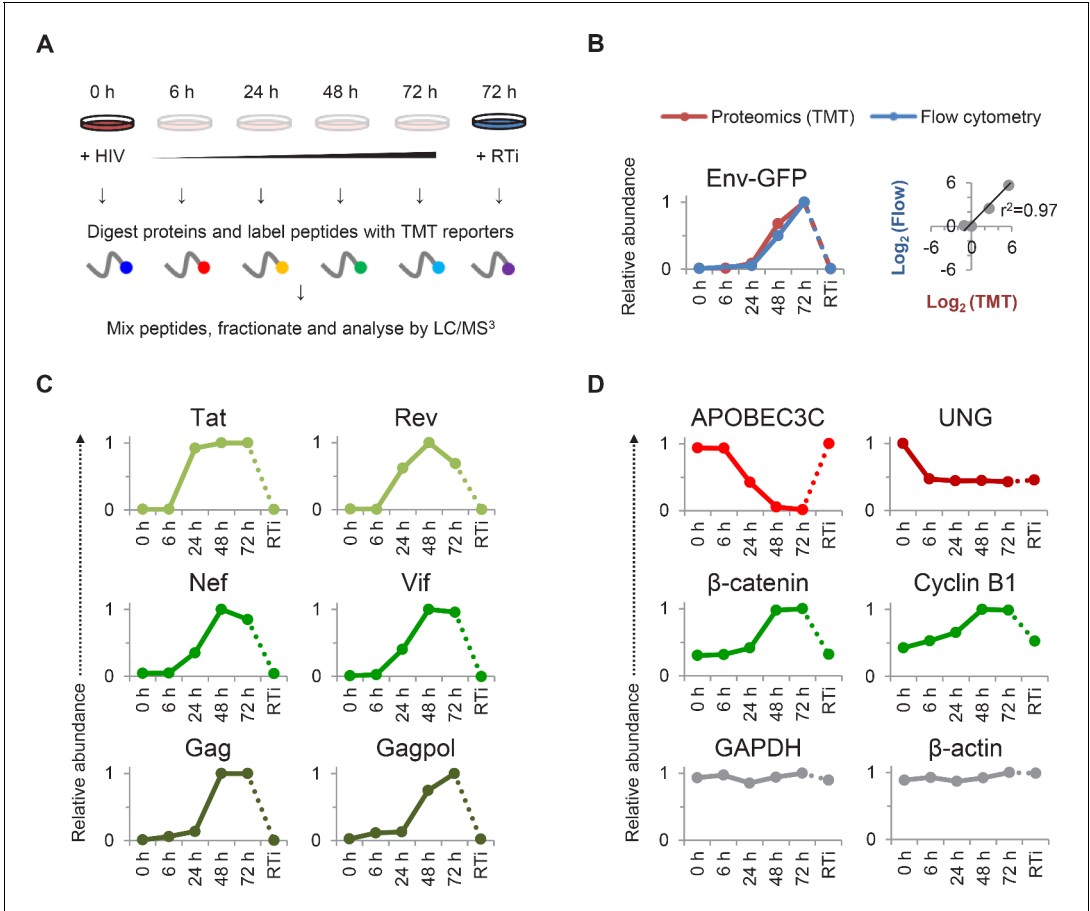

**Figure 1.** TMT-based proteomic time course analysis of HIV-infected cells. (**A**) Workflow of 6-plex TMT-based whole cell proteomic time course experiment. CEM-T4 T cells were infected with NL4-3-dE-EGFP HIV at an MOI of 10. In subsequent figures timepoints 1–5 show protein abundance 0, 6, 24, 48 and 72 hr after HIV infection (where 0 hr = uninfected cells) and timepoint 6 shows protein abundance 72 hr after HIV infection in the presence of reverse transcriptase inhibitors (RTi). (**B**) Comparison of temporal profiles of Env-GFP obtained by proteomic (TMT) versus flow cytometric quantitation. Cells from (**A**) were analysed by flow cytometry. Relative abundance is expressed as a fraction of maximum TMT reporter ion or fluorescence intensity. For linear regression, $\log_2$ (fold change in protein abundance compared with uninfected cells) is shown. (**C–D**) Temporal profiles of viral proteins (**C**) and previously reported HIV targets (**D**). GAPDH and β-actin are included as controls. Relative abundance is expressed as a fraction of maximum TMT reporter ion intensity.

The following source data and figure supplements are available for figure 1:

**Source data 1.** Interactive spreadsheet of TMT time course data.

**Figure supplement 1.** Additional temporal profiles and comparison with plasma membrane profiling.

**Figure supplement 2.** Gene set enrichment analysis of HIV infection.

We observed a tight correlation between levels of Env-GFP expression determined by mass spectrometry and flow cytometry ($r^2$ = 0.97) (*Figure 1B*). As expected, the well characterised HIV cell surface targets downregulated in our plasma membrane proteomic analysis were also depleted in our whole cell proteomic analysis (*Figure 1—figure supplement 1A*). The magnitude of effect was generally greater in the plasma membrane proteomic analysis (*Figure 1—figure supplement 1A–B*), suggesting that regulation of cell surface proteins by redistribution or sequestration is an important feature of this system.

We detected gene products from 7/9 HIV-1 open reading frames (ORFs; *Figure 1B–C*). As previously reported, expression of regulatory proteins (Tat and Rev) from Rev-independent completely spliced mRNA transcripts occurred earlier in viral replication than expression of structural proteins from Rev-dependent unspliced (Gag and Gagpol) and partially spliced (Env) mRNA transcripts (*Karn and Stoltzfus, 2012*; *Pollard and Malim, 1998*), with Rev expression lagging Tat in our experiment. Vif and Nef showed intermediate temporal profiles (*Figure 1C*), with progressively increasing Nef expression from 24–48 hr inversely correlating with downregulation of CD4 and HLA-A (*Figure 1—figure supplement 1A*). Finally, we saw an increase in plasma membrane VSVg levels immediately after infection (reflecting fusion of incoming virions), followed by a rapid decline (*Figure 1—figure supplement 1C*).

Compared with numerous cell surface targets (*Haller et al., 2014*; *Matheson et al., 2015*), relatively few intracellular proteins depleted by HIV accessory proteins have been described. Nonetheless, we confirmed downregulation of the Vif target APOBEC3C (*Smith and Pathak, 2010*) and the Vpr target UNG (*Schrofelbauer et al., 2005*) (*Figure 1D*). The temporal pattern of UNG depletion was distinct from that of other accessory protein substrates, including APOBEC3C, with degradation seen as early as 6 hr post-infection, and preserved in the presence of reverse transcriptase inhibitors. This is likely to reflect the high abundance of Vpr packaged within incoming viral particles (*Lu et al., 1993*; *Paxton et al., 1993*), abrogating the need for de novo protein synthesis. As well as recruiting substrates for degradation, Vpu increases β-catenin levels by sequestering the ß-TrCP substrate-recognition unit of the SCF$^{β-TrCP}$ E3 ubiquitin ligase complex (*Besnard-Guerin et al., 2004*). In addition, HIV infection causes cell cycle arrest at G2/M (*Jowett et al., 1995*), a point in the cell cycle associated with upregulation of cyclin B1 (*Norbury and Nurse, 1992*). Accordingly, we observed progressive accumulation of both β-catenin and cyclin B1 (*Figure 1D*).

## Temporal clustering of cellular proteins modulated by HIV

Gene Set Enrichment Analysis (GSEA) revealed time-dependent perturbation of multiple cellular processes and pathways during HIV infection (*Figure 1—figure supplement 2A–B*), with protein-level changes generally supporting earlier transcriptome-level data. For example, genes associated with lipid metabolism (*Figure 1—figure supplement 2C*) are induced by expression of Nef (*Shrivastava et al., 2016*; *van 't Wout et al., 2005*), whereas genes associated with RNA processing (*Figure 1—figure supplement 2D*) are suppressed in HIV-infected cells (*Chang et al., 2011*; *Sherrill-Mix et al., 2015*).

To facilitate data mining and identify specific host factors regulated by HIV infection, we classified cellular proteins according to their patterns of temporal expression (*Figure 2A*). We observed 4 main clusters: (#1) 29 proteins downregulated late in infection, rescued in the presence of reverse transcriptase inhibitors; (#2) 59 proteins downregulated earlier in infection, incompletely rescued in the presence of reverse transcriptase inhibitors; (#3) 29 proteins progressively upregulated during infection, abolished in the presence of reverse transcriptase inhibitors; and (#4) 49 proteins progressively upregulated during infection, even in the presence of reverse transcriptase inhibitors (*Figure 2B*). We validated protein downregulation (clusters #1 and #2) and upregulation (clusters #3 and #4) in an independent infection time course experiment, using stable isotope labelling with amino acids in cell culture (SILAC) as an alternative quantitative proteomic approach (*Figure 2C* and *Figure 2—figure supplement 1A*). Details of all proteins in clusters #1–4, including validation time course data, are available in *Figure 2—source data 1*.

Distinct patterns of temporal regulation imply different mechanisms and biological significance. The Nef, Vpu and Vif accessory protein targets CD4, SNAT1, APOBEC3C are found in cluster #1, where progressive downregulation and rescue by reverse transcriptase inhibitors suggest dependence on de novo viral protein synthesis (compare *Figure 2B* with *Figure 1D*, top left panel, and *Figure 1—figure supplement 1A*). Upregulation of proteins in cluster #3 is also likely to require de

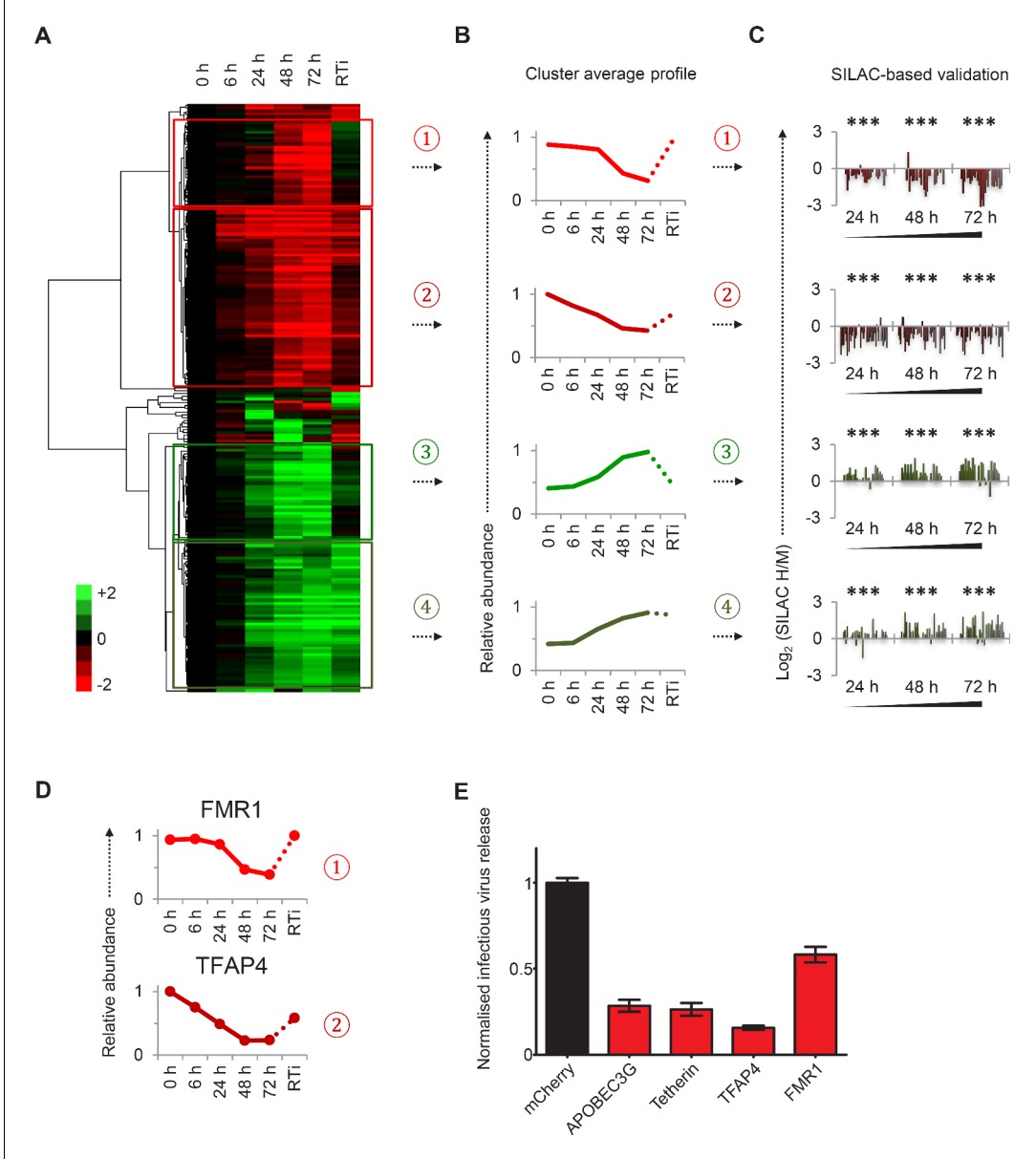

**Figure 2.** Identification and SILAC-based proteomic validation of novel HIV targets. (**A**) Hierarchical cluster analysis of temporal profiles of proteins regulated by HIV. The heatmap shows $\log_2$ (fold change in protein abundance compared with uninfected cells) and clusters #1–4 are indicated. (**B**) Average temporal profiles of proteins in clusters #1–4. Relative abundance is expressed as a fraction of maximum TMT reporter ion intensity. (**C**) SILAC-based validation of novel HIV targets. CEM-T4 T cells pre-labelled with heavy amino acids were infected with NL4-3-dE-EGFP HIV at an MOI of 10, and control cells pre-labelled with medium amino acids were mock-infected without virus. Aliquots of HIV-infected (heavy; H) and mock (medium; M) cells were harvested sequentially at the indicated timepoints and subjected to SILAC-based whole cell proteomic analysis (*Figure 2—figure supplement 1A*). $\log_2$ (H/M protein abundance) at 24, 48 and 72 hr is shown for proteins from clusters #1–4. ***p value<0.001. (**D**) Temporal profiles of novel HIV-1 targets FMR1 and TFAP4. Relative abundance is expressed as a fraction of maximum TMT reporter ion intensity. (**E**) Antagonism of HIV production by FMR1 and TFAP4. 293T cells were co-transfected with pNL4-3-dE-EGFP/pMD.G and either mCherry or the indicated cellular protein. 48 hr culture supernatants were assayed for infectious virus by infection of HeLa cells. Well-characterised restriction factors APOBEC3G and tetherin were included as controls, and infectious virus release normalised compared with mCherry. Mean values and standard errors are shown from at least 4 replicates. All four proteins significantly reduced viral release compared with mCherry in an ANOVA analysis with Bonferroni post-test, p values<0.001.

The following source data and figure supplement are available for figure 2:

**Source data 1.** Clusters #1–4 summary proteomic data.

*Figure 2 continued on next page*

*Figure 2 continued*

**Figure supplement 1.** Workflow of SILAC-based proteomic time course experiment, functional analysis of clusters #1–4 and prediction of novel Vpr targets.

novo viral protein synthesis, and the indirect Vpu target β-catenin is found in this cluster (compare *Figure 2B* with *Figure 1D*, middle left panel). Conversely, reverse transcriptase inhibitor-independent regulation of proteins in clusters #2 and #4 implies a cellular response to HIV infection, or a direct effect of viral proteins in incoming virions, and the Vpr target UNG is found in cluster #2 (compare *Figure 2B* with *Figure 1D*, top right panel). As well as mechanistic differences, analysis using the Database for Annotation, Visualisation and Integrated Discovery (DAVID) revealed that clusters #1–4 contained proteins associated with distinct biological functions and processes (*Figure 2—figure supplement 1B*). Whilst accumulation of some proteins in cluster #3 may be secondary to G2/M cell cycle arrest, other changes are unlikely to reflect the interferon (IFN) or unfolded protein responses, because we did not see accumulation of either the highly IFN-inducible protein ISG15 (*Figure 1—figure supplement 1D*) or proteins associated with ER stress (*Figure 1—figure supplement 1E*).

## Regulation of resistance/restriction factors and candidate accessory protein targets

Restriction factors are cellular proteins whose primary biological activity is antiviral, and which are induced by IFN or viral infection, antagonized by viral proteins, and show genetic evidence of positive selection (*Duggal and Emerman, 2012*). Proteins which reduce permissivity for viral infection, but fail to meet strict criteria for restriction factors, may be classified as resistance factors (*Goujon et al., 2013*). Restriction and resistance factors are characteristically increased (cellular response) or decreased (viral antagonism) during viral infection. Our temporal proteomic approach therefore identifies unsuspected viral regulation of known resistance/restriction factors, and provides a strategy for the discovery of novel host antiviral factors. Accordingly, we found the actin regulatory proteins gelsolin and CAPG (both cluster #4) to be strongly induced during HIV infection (*Figure 1—figure supplement 1F*). Actin cytoskeletal remodeling is required for virological synapse formation and cell-cell transmission of HIV (*Jolly et al., 2004*), and gelsolin levels have been reported to control early HIV infection in macrophages (*Garcia-Exposito et al., 2013*). In contrast, we discovered marked depletion of FMR1 (cluster #1, *Figure 2D*, **upper panel**) and TFAP4 (cluster #2, *Figure 2D*, **lower panel**) during HIV infection. Both these proteins reduce production of infectious HIV virus (*Figure 2E*) (*Imai and Okamoto, 2006*; *Pan et al., 2009*), but their distinct patterns of temporal expression suggest different mechanisms of viral regulation. We predict that other regulated proteins in clusters #1–4, without known roles in HIV infection, will also represent novel cellular resistance/restriction factors.

To test the utility of our approach, we focussed on proteins in cluster #1 highlighted in our functional analysis (*Figure 3A–B* and *Figure 2—figure supplement 1B*). First, three members of the deoxynucleotide triphosphate (dNTP) biosynthetic pathway were depleted during HIV infection: thymidylate synthetase (TYMS), which catalyses the methylation of deoxyuridylate (dUMP) to deoxythymidylate (dTMP); and two subunits of ribonucleotide reductase (RNR), RRM1 and RRM2, which catalyses the formation of deoxyribonucleotides from ribonucleotides (*Figure 3B*, **left panels**). HIV replication is tightly regulated by dNTP availability, and SAMHD1 (which tends to oppose the effects of RNR) is a well-described HIV restriction factor (*Ayinde et al., 2012*; *Baldauf et al., 2012*; *Hrecka et al., 2011*; *Laguette et al., 2011*; *Lahouassa et al., 2012*; *Taylor et al., 2015*). Second, and most strikingly, all detected members of the B56 family of protein phosphatase 2 A (PP2A) regulatory subunits (PPP2R5A, C, D and E) were profoundly depleted by HIV (*Figure 3B*, **right panels**). These subunits determine the specificity and localisation of PP2A holoenzymes (PP2A-B56), a ubiquitous family of heterotrimeric serine-threonine phosphatases with critical roles in many aspects of cellular physiology (*McCright et al., 1996*). We confirmed downregulation of RRM2, PPP2R5A and PPP2R5D by immunoblot (*Figure 3—figure supplement 1*). Because cluster #1 also contained APOBEC3C, and intracellular proteins in this cluster (including PPP2R5 subunits) show near-identical

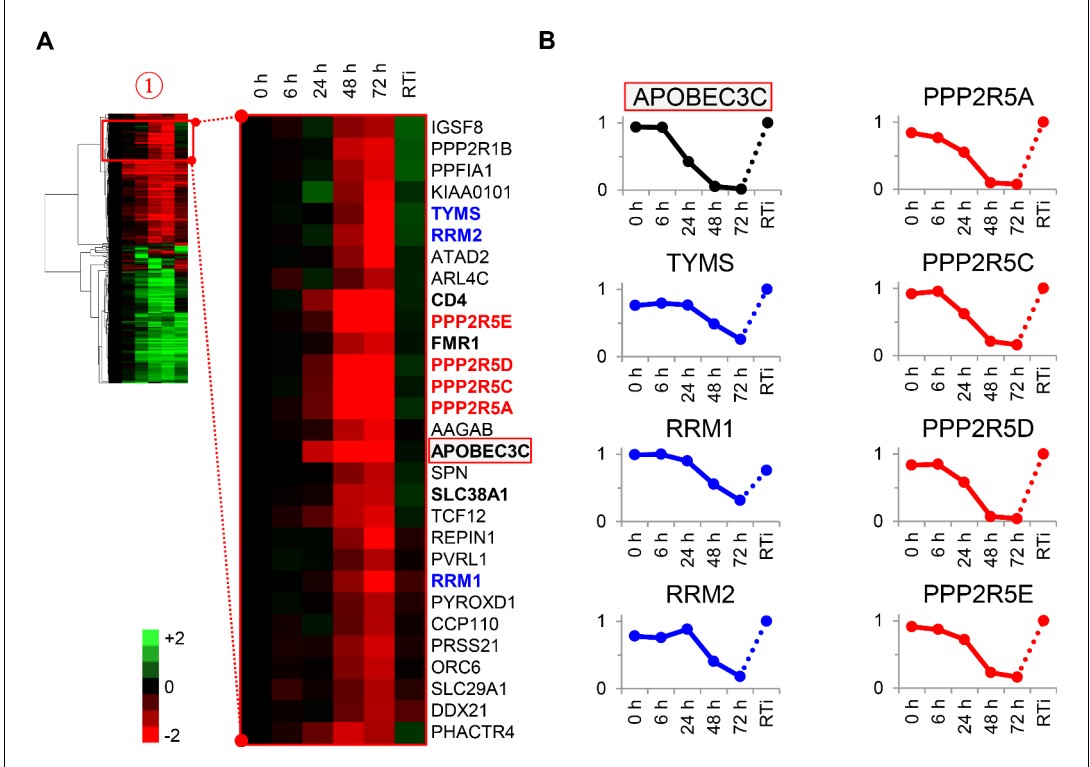

**Figure 3.** Cellular proteins progressively downregulated by HIV infection. (**A**) Enlargement of cluster #1 from hierarchical cluster analysis (*Figure 2A*). The heatmap shows $\log_2$ (fold change in protein abundance compared with uninfected cells). Enzymes associated with deoxynucleotide metabolism (blue) and B56 family regulatory subunits of serine/threonine protein phosphatase PP2A (red) are highlighted, along with known Vif target APOBEC3C (boxed) and other proteins of interest (bold). (**B**) Temporal profiles of enzymes associated with deoxynucleotide metabolism (blue) and B56 family regulatory subunits of serine/threonine protein phosphatase PP2A (red). Relative abundance is expressed as a fraction of maximum TMT reporter ion intensity, and the temporal profile of APOBEC3C is shown for comparison.

The following figure supplement is available for figure 3:

**Figure supplement 1.** Immunoblot validation of novel HIV targets.

patterns of temporal expression, we hypothesised that some or all of these proteins might be novel Vif targets.

## Systematic multiplex proteomic analysis of Vif targets

To examine this hypothesis and systematically identify novel Vif targets, we performed a 3-way proteomic comparison of mock-infected cells and cells infected with wildtype (WT) or Vif-deficient (ΔVif) HIV viruses (*Figure 4—figure supplement 1A*). To complement our high MOI time course experiment, we used an MOI of 1.5, resulting in approximately 75% productive infection and (typically) one or two copies of the viral genome per cell (*Figure 4—figure supplement 1B*). We exploited 10-plex TMT labelling to analyse samples in triplicate at a single timepoint 48 hr post-infection, with the resulting statistical power compensating for the reduced magnitude of changes due to the presence of bystander (uninfected) cells. As expected, Vif protein was only detected in WT infection, but levels of other viral proteins were equivalent (*Figure 4A*).

Vif-independent HIV targets such as tetherin (Vpu substrate), SNAT1 (Vpu substrate), CD4 (Nef/Vpu substrate) and UNG (Vpr substrate) were depleted in cells infected with both WT (*Figure 4A*, **left panel**) and ΔVif (*Figure 4A*, **middle panel**) viruses, with no difference in abundance in the presence or absence of Vif (*Figure 4A*, **right panel**). Conversely, known Vif targets APOBEC3C, APOBEC3G and APOBEC3D were all decreased by WT but not ΔVif HIV infection (*Figure 4A*), and APOBEC3B, which is resistant to Vif (*Doehle et al., 2005*), was unchanged across all conditions. In

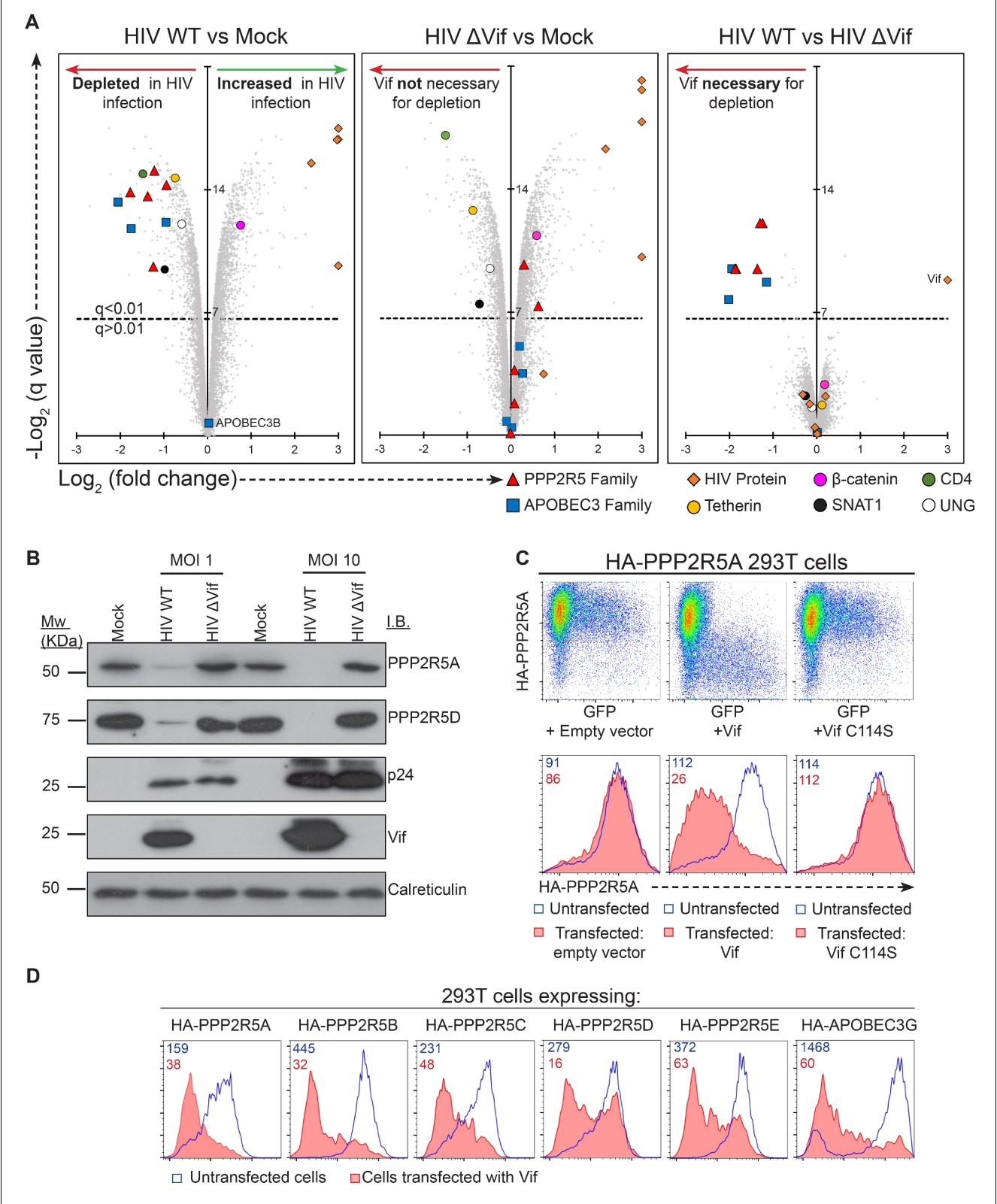

**Figure 4.** Vif-mediated depletion of PP2A-B56 family members PPP2R5A-E. (**A**) Proteomic analysis of CEM-T4 T cells infected with WT and ΔVif HIV. Cells were infected with NL4-3-ΔE-EGFP viruses at an MOI of 1.5, and harvested 48 hr post-infection. Scatterplots display pairwise comparisons between WT, ΔVif and mock-infected cells. Each point represents a single protein, plotted by its log2 (fold change in abundance) versus the statistical significance of that change. q values were determined using Limma with Benjamini-Hochberg adjustment for multiple testing, with increasing −log2 (q

*Figure 4 continued on next page*

Figure 4 continued

value) indicating increasing significance. Points above the dotted line change with a q value < 0.01. HIV proteins and host proteins of interest are highlighted with different symbols (see key). (B) Depletion of PPP2R5A and PPP2R5D during HIV infection. CEM-T4 T-cells were infected with NL4-3-dE-EGFP WT and ΔVif viruses at an MOI of 1 or 10 and analysed by immunoblot (IB) 48 hr post-infection. p24 (capsid), Vif and calreticulin are included as controls. (C) Depletion of exogenous PPP2R5A by Vif. 293T cells stably expressing HA-PPP2R5A were co-transfected with GFP plus empty vector, NL4-3 Vif or NL4-3 Vif with a single amino acid mutation C114S and analysed by intracellular flow cytometry for HA 36 hr post-transfection. Histograms show GFP positive (transfected, red shading) and negative (untransfected, blue line) cells. Median fluorescence intensity (MFI) values are shown for GFP positive (red) and negative (blue) cells. (D) Depletion of PPP2R5A-E family members by Vif. 293T cells stably expressing HA-tagged PPP2R5A-E or APOBEC3G were co-transfected with GFP plus NL4-3 Vif expression vectors, and intracellular HA staining quantitated by flow cytometry 36 hr post transfection. Histograms show GFP positive (transfected, red shading) and negative (untransfected, blue line) cells. MFI values are shown for GFP positive (red) and negative (blue) cells.

The following figure supplements are available for figure 4:

**Figure supplement 1.** Workflow and controls for single timepoint proteomic/phosphoproteomic experiment.

**Figure supplement 2.** Depletion of endogenous PPP2R5D during HIV infection of primary cells.

addition to APOBEC family members, all five PP2A-B56 regulatory subunits PPP2R5A-E were depleted in the presence of Vif (*Figure 4A*). Vif-dependent degradation of PPP2R5 subunits was confirmed by immunoblot of HIV-infected CEM-T4 T cells (PPP2R5A and PPP2R5D; *Figure 4B*) and intracellular flow cytometry of HIV-infected CEM-T4 and primary human CD4[+] T cells (PPP2R5D; *Figure 4—figure supplement 2A–B*).

## CUL5-dependent proteasomal degradation of PPP2R5 subunits

To test whether degradation of PPP2R5 subunits by Vif was post-translational, we expressed HA-tagged PPP2R5A in 293T cells. Transfection of Vif caused a marked loss of intracellular HA staining (*Figure 4C*, **middle panels**), and the same effect was also seen in cells expressing all other PPP2R5 subunits or APOBEC3G (*Figure 4D*). Degradation of APOBEC family members by Vif is mediated by recruitment of a cullin-5 (CUL5) E3 ubiquitin ligase complex, resulting in substrate-specific ubiquitination and proteasomal degradation (*Malim and Bieniasz, 2012*). We therefore predicted that depletion of PPP2R5 subunits would exploit the same pathway.

Accordingly, we found that the Vif C114S mutant, which is unable to recruit CUL5 (*Bergeron et al., 2010*), was defective for PPP2R5A degradation (*Figure 4C*, **right panels**), and PPP2R5A degradation by wildtype Vif was rescued in the presence of the proteasome inhibitor bortezomib (*Figure 5A*). A similar rescue of PPP2R5B was seen when Vif was co-transfected with dominant negative, but not wildtype, CUL5 (*Figure 5B*), and knockdown of other cellular components of the CUL5 E3 ligase complex recruited by Vif (EloB, EloC and CBFβ) (*Jager et al., 2012*; *Malim and Bieniasz, 2012*) rescued both PPP2R5B and APOBEC3G from degradation, with the magnitude of rescue similar for both substrates (*Figure 5C*).

Consistent with a protein-level interaction between Vif and PPP2R5 subunits, we observed co-immunoprecipitation of FLAG-tagged Vif with HA-tagged and endogenous PPP2R5D in 293T cells (*Figure 5—figure supplement 1A*), and co-immunoprecipitation of untagged Vif with endogenous PPP2R5D in CEM-T4 T cells infected with HIV (*Figure 5—figure supplement 1B*). As in 293T cells transfected with Vif, PPP2R5 subunit depletion in CEM-T4 T cells infected with HIV was abolished in the presence of bortezomib (*Figure 5—figure supplement 2A–B*). Finally, we confirmed using cycloheximide chase (*Figure 5—figure supplement 2C–D*) and [35S]methionine/[35S]cysteine metabolic labelling/pulse-chase (*Figure 5—figure supplement 3*) analyses that degradation of PPP2R5D was accelerated in the presence of Vif in HIV-infected CEM-T4 T cells. Vif is therefore both necessary and sufficient for degradation of PPP2R5 subunits, and employs the same cellular machinery required for degradation of APOBEC family members.

## Remodelling of the cellular phosphoproteome by HIV infection

The substrate specificity of the PP2A phosphatase holoenzyme is determined by binding of its regulatory subunits (*Yang and Phiel, 2010*). To identify the phenotypic consequences of Vif-mediated

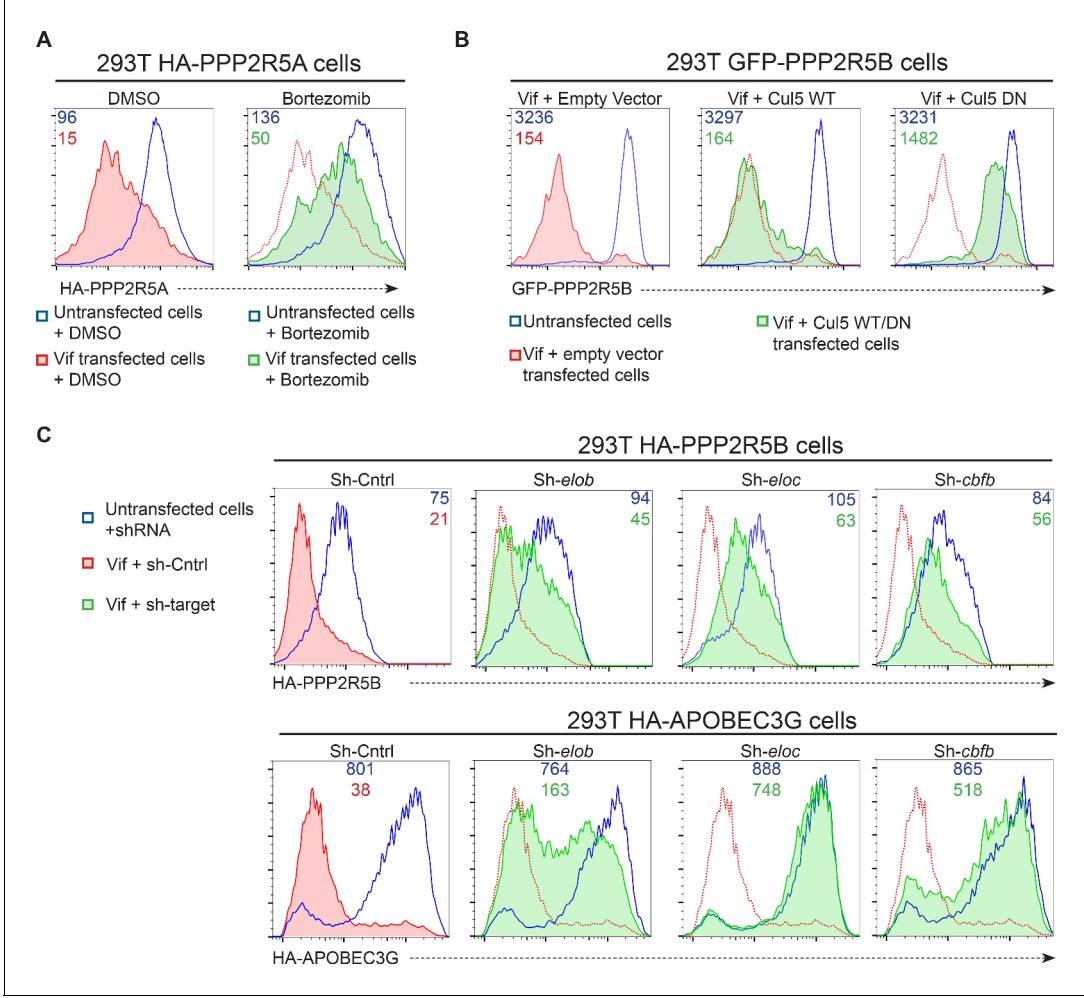

**Figure 5.** Mechanism of Vif-mediated degradation of PPP2R5A-E subunits. (**A**) Proteasomal degradation. 293T cells stably expressing HA-PPP2R5A were transfected with NL4-3 Vif in the presence of DMSO (control) or the proteasome inhibitor bortezomib and analysed by intracellular flow cytometry for HA. (**B**) CUL5-dependent degradation. 293T cells stably expressing GFP-PPP2R5B were co-transfected with NL4-3 Vif plus empty vector, wildtype cullin-5 (CUL5 WT) or a dominant negative cullin-5 mutant (CUL5 DN) and analysed by flow cytometry for GFP. (**C**) CUL5 complex-dependent degradation. 293T cells stably expressing HA-PPP2R5B (upper panels) or HA-APOBEC3G (lower panels) were transduced with the indicated shRNA. Cells were then transfected with NL4-3 Vif and analysed by intracellular flow cytometry for HA. Green/red shading shows Vif-transfected cells in the indicated shRNA background. Red lines showing HA staining in cells transduced with control shRNA are included in each panel for reference. In all experiments, cells were analysed 36 hr post-transfection, and transfected cells determined by co-transfection with GFP (**A** and **C**) or mCherry (**B**). MFI values are shown for transfected (red/green) and untransfected (blue) cells.

The following figure supplements are available for figure 5:

**Figure supplement 1.** Co-immunoprecipitation of Vif and PPP2R5D.

**Figure supplement 2.** Time course analysis of endogenous PPP2R5D during HIV infection of T cells.

**Figure supplement 3.** Pulse-chase analysis of endogenous PPP2R5D during HIV infection of T cells.

PPP2R5A-E subunit depletion, we used titanium dioxide-based phosphopeptide enrichment and 10-plex TMT labelling to analyse total phosphoproteomes of the mock-, WT and ΔVif virus-infected cells described in *Figure 4A* and *Figure 4—figure supplement 1A–B*. In total, we quantitated 8631 phosphopeptides from 2767 proteins (*Figure 6—source data 1*). Phosphopeptide abundance was normalized to total protein abundance determined from the whole cell proteomic analysis, allowing

differential phosphorylation to be distinguished from altered protein expression (*Wu et al., 2011*). HIV infection resulted in marked remodelling of the cellular phosphoproteome (*Figure 6—figure supplement 1A*), and analysis using the Database for Annotation, Visualisation and Integrated Discovery (DAVID) revealed enhanced phosphorylation of proteins associated with cell cycle regulation and activation of the DNA damage response (*Figure 6—figure supplement 1B*). To isolate those changes which specifically resulted from Vif-mediated PPP2R5A-E subunit depletion, we focused on differences between cells infected with WT and ΔVif viruses.

Remarkably, compared with the small number of protein-level changes in the presence or absence of Vif (specifically, APOBEC and PPP2R5 family members; *Figure 4A*, **right panel** and *Figure 6A*, **left panel**), we saw striking Vif-dependent changes in the phosphoproteome (*Figure 6A*, **right panel**). Furthermore, as predicted for antagonism of a phosphatase, almost all changes represented increases in phosphopeptide abundance, indicating increased protein phosphorylation (with a total of 238 peptides from 192 proteins showing abundance changes of >2 fold with a q value of <0.01). To confirm that the observed changes resulted from PP2A antagonism, we compared our Vif-dependent changes in protein phosphorylation with published alterations to the phosphoproteome of HeLa cells following treatment with the PP2A inhibitor okadaic acid (*Kauko et al., 2015*). Despite the different cell types and treatments, a highly significant correlation was found between our observed Vif-dependent changes in HIV-infected cells and the published changes resulting from okadaic acid treatment (*Figure 6B* and *Figure 6—figure supplement 1D*).

To identify individual kinases with enhanced activity in the presence of Vif-dependent PPP2R5A-E subunit depletion, we interrogated our data using PhosFate (http://phosfate.com/), which infers kinase activity from quantitative phosphoproteomic data by examining the coordinated regulation of known phosphosites (*Ochoa et al., 2016*). We found marked activation of aurora kinase A (AURKA) and B (AURKB) in cells infected with WT but not ΔVif viruses (*Figure 6—figure supplement 1C*), and confirmed this observation by comparing phosphorylation of sites listed on the PhosphoSitePlus (http://www.phosphosite.org/) kinase substrate database (*Hornbeck et al., 2015*) between WT and ΔVif virus infections (*Figure 6C*). Next, we compared Vif-dependent changes in protein phosphorylation with published alterations to the phosphoproteome of HeLa cells following treatment with the aurora kinase inhibitors MLN8054 (*Figure 6D*, **upper panel**, and *Figure 6—figure supplement 1E*) and AZD1152/ZM447439 (AZDZM; *Figure 6—figure supplement 1E*) (*Kettenbach et al., 2011*). As expected, we found a significant inverse correlation between our Vif-dependent changes in HIV-infected cells and the published changes resulting from aurora kinase inhibition, whereas no such correlation was seen for control datasets from the same study employing DMSO or the PLK1-3 inhibitor BI2536 (*Figure 6D*, **lower panel**, and *Figure 6—figure supplement 1E*).

Finally, to fully characterize the behavior of these kinases in our dataset, we manually curated the literature for substrates of aurora kinases, including PLK1 as a negative control for Vif-specific effects (*Figure 6—source data 2*). PLK1 protein abundance was upregulated in both WT and ΔVif infections, with enhanced phosphorylation of kinase-specific phosphosites, but no difference between WT and ΔVif viruses (*Figure 6E*, **left panels**, and *Figure 1—figure supplement 1G*). By contrast, whilst the aurora kinases were also upregulated equally in WT and ΔVif infections, increased phosphorylation of kinase-specific phosphosites was only seen in the presence of Vif (*Figure 6E*, **middle and right panels**, and *Figure 1—figure supplement 1G*). Depletion of PPP2R5A-E subunits by Vif is therefore responsible for the selective amplification of aurora kinase activity in HIV-infected T cells.

## Depletion of PPP2R5A-E subunits by phylogenetically diverse lentiviral Vif variants

The *vif* gene is found in all primate lentiviral lineages, and in most of the extant non-primate lineages. We therefore assembled a panel of *vif* genes from diverse primate and non-primate lentiviruses (*Figure 7A* and *Figure 7—figure supplement 1*), including 14 *vif* variants from HIV-1 clades A-F and 6 *vif* variants from SIVcpz and SIVgor of chimpanzees and gorillas, the most closely related viruses to HIV. Multiple *vif* variants from two other primate lentiviral lineages were also represented: SIVsmm of sooty mangabeys, and the viruses that resulted from cross species transmission of SIVsmm, HIV-2 and SIVmac; and SIVagm of African green monkeys. Finally, a non-primate lentivirus *vif* variant was included, from a small ruminant lentivirus (SRLV, or maedi-visna virus) isolated from sheep (*Sargan et al., 1991*).

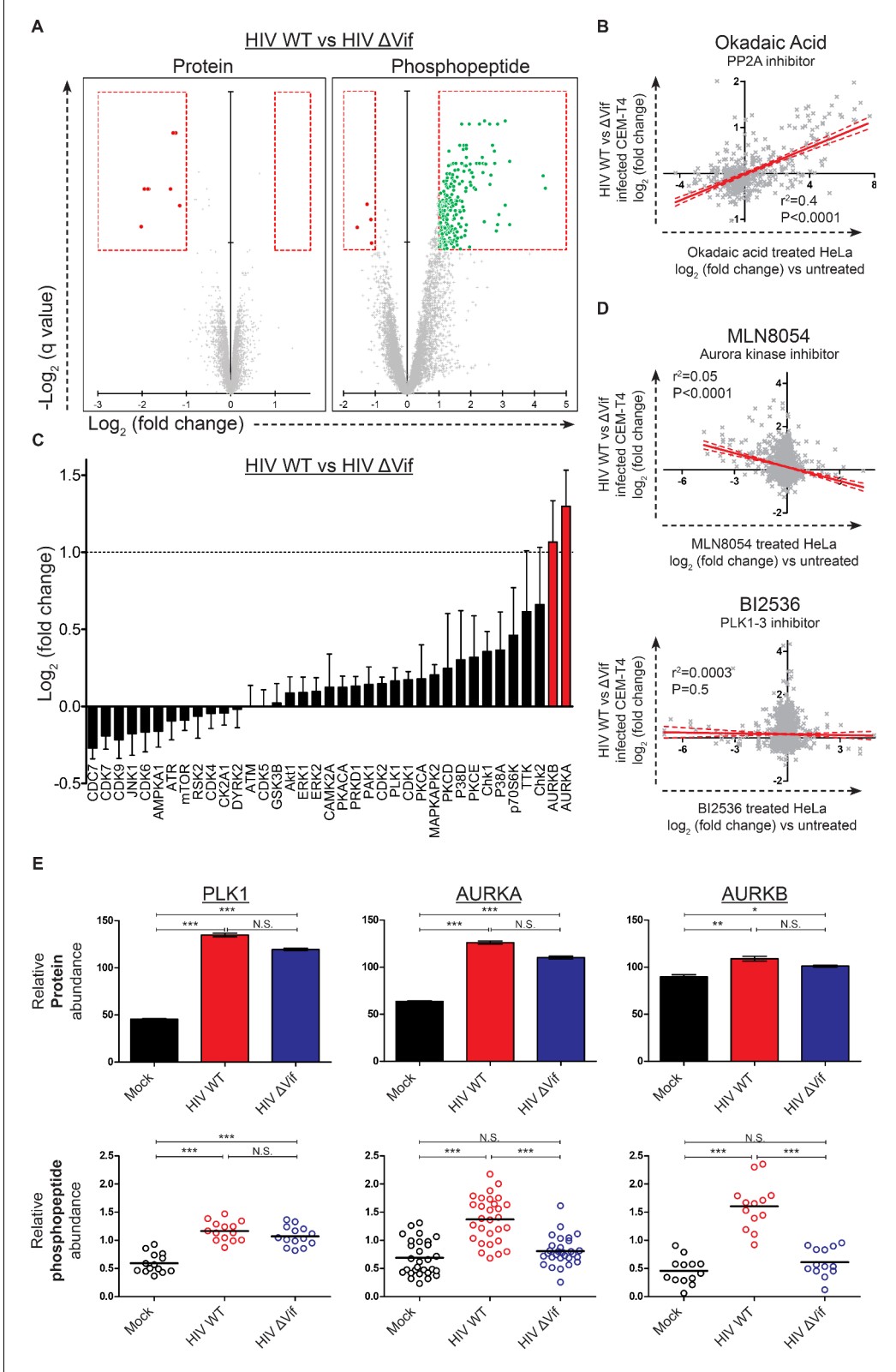

**Figure 6.** Global phosphoproteomic analysis of cells infected with WT or ΔVif HIV. (**A**) Vif-dependent changes in peptide and phosphopeptide abundance. CEM-T4 T cells from *Figure 4A* and *Figure 4—figure supplement 1A* were subjected to TMT-based phosphoproteomic analysis. Scatterplots display differences in protein (**left panel**, as in *Figure 4A*, **right panel**) and phosphopeptide abundance (**right panel**) between WT and ΔVif-infected cells. Each point represents a single protein or phosphopeptide, plotted by its $\log_2$ (fold change in abundance) versus the statistical

*Figure 6 continued on next page*

*Figure 6 continued*

significance of that change. q values were determined using Limma with Benjamini-Hochberg adjustment for multiple testing, with increasing −log2 (q value) indicating increasing significance. Proteins and phosphopeptides downregulated (red) or upregulated (green) with a fold change > 2 and q value < 0.01 are highlighted. (B) Comparison of changes in phosphopeptide abundance between WT and ΔVif-infected CEM-T4 T cells with previously published data for okadaic acid-treated HeLa cells (*Kauko et al., 2015*). Lines show linear correlation with associated 95% confidence areas, $r^2$ values and p values of a non-zero correlation. (C) Analysis of changes in phosphopeptide abundance between WT and ΔVif-infected cells CEM-T4 T cells using the PhosphoSitePlus kinase-substrate database. Bars show $\log_2$ (fold change in phosphopeptide abundance) for peptides spanning known kinase substrate sites. Error bars show the standard error of the mean. (D) Comparison of changes in phosphopeptide abundance between WT and ΔVif-infected CEM-T4 T cells with previously published data for kinase inhibitor-treated HeLa cells (*Kettenbach et al., 2011*). At low concentrations, MLN8054 is a selective AURKA inhibitor, but at 5 μM (as shown) reduced activity of AURKB and PLK1 is also observed. Lines show linear correlation with associated 95% confidence areas, $r^2$ values and p values of a non-zero correlation. (E) Vif-specific hyperphosphorylation of aurora kinase substrates. Protein abundances of PLK1, AURKA and AURKB were compared with normalised abundances of manually curated phosphopeptides targeted by the respective kinases. Abundances of kinase proteins were compared using Limma with Benjamini-Hochberg adjustment for multiple testing. Abundances of target phosphopeptides were compared by Repeated Measures ANOVA with Bonferroni post-test. N.S., p value>0.05; *p value<0.05; **p value<0.01; ***p value<0.001.

The following source data and figure supplement are available for figure 6:

**Source data 1.** Single timepoint phosphoproteomic data.
**Source data 2.** Previously reported AURKA, AURKB and PLK1 targets.
**Figure supplement 1.** Further phosphoproteomic analysis.

Vif variants were tested by transfection of 293T cells stably expressing HA-tagged PPP2R5 subunits, with the proportion of HA-tagged protein degraded in transfected cells quantitated by intracellular flow cytometry. All HIV-1 variants tested were able to degrade HA-PPP2R5A, but the magnitude of effect was variable (*Figure 7—figure supplement 2A*). We therefore screened a diverse selection of Vif variants for degradation of different PPP2R5 subunits (*Figure 7—figure supplement 2B*). The ability to deplete PPP2R5 subunits was conserved across all PPP2R5A-E/Vif combinations, but most marked for PPP2R5B. We therefore tested our entire panel of Vif variants for depletion of PPP2R5B, and found strong and consistent degradation (*Figure 7B* and *Figure 7—figure supplement 2C*).

Finally, we focused specifically on the distantly related SRLV and NL4-3 (HIV-1) Vif variants. Vif-dependent antagonism of APOBEC proteins shows lineage-specificity, and SRLV Vif is unable to antagonize human APOBEC3G (*Larue et al., 2010*). Nonetheless, despite only sharing 15% amino acid identity with NL4-3 Vif (*Figure 7—figure supplement 1*), SRLV Vif was still able to associate with (*Figure 7—figure supplement 3A*) and efficiently degrade human PPP2R5 subunits (*Figure 7C*). Whilst Vif variants from primate lentiviruses (including HIV-1) require CBFβ to enable proper protein folding, stability and interaction with the CUL5 E3 ligase complex (*Fribourgh et al., 2014*; *Kim et al., 2013*; *Miyagi et al., 2014*; *Salter et al., 2012*) and mediate APOBEC depletion (*Hultquist et al., 2012*; *Jager et al., 2012*; *Zhang et al., 2012*), Vif variants from non-primate lentiviruses (including SRLV) neither interact with CBFβ (*Ai et al., 2014*; *Kane et al., 2015*; *Yoshikawa et al., 2016*; *Zhang et al., 2014*) nor require CBFβ to antagonize their cognate APOBEC proteins (*Ai et al., 2014*; *Kane et al., 2015*). As with APOBEC proteins, we found CBFβ but not EloB to be dispensable for degradation of HA-PPP2R5E by SRLV Vif (*Figure 7—figure supplement 3B*).

## Discussion

In this study, we provide a comprehensive description of temporal changes in >6500 viral and cellular proteins during HIV infection. Our data confirm known HIV targets, and identify many more proteins regulated by infection. Compared with other studies (*Supplementary file 1*), we achieve a step-change in depth of proteomic coverage, and by utilising multiplex TMT-based quantitation, we facilitate high-resolution time-based analysis. To generate a cell surface proteomic map of HIV infection, we previously employed selective aminooxy-biotinylation of sialylated glycoproteins (Plasma

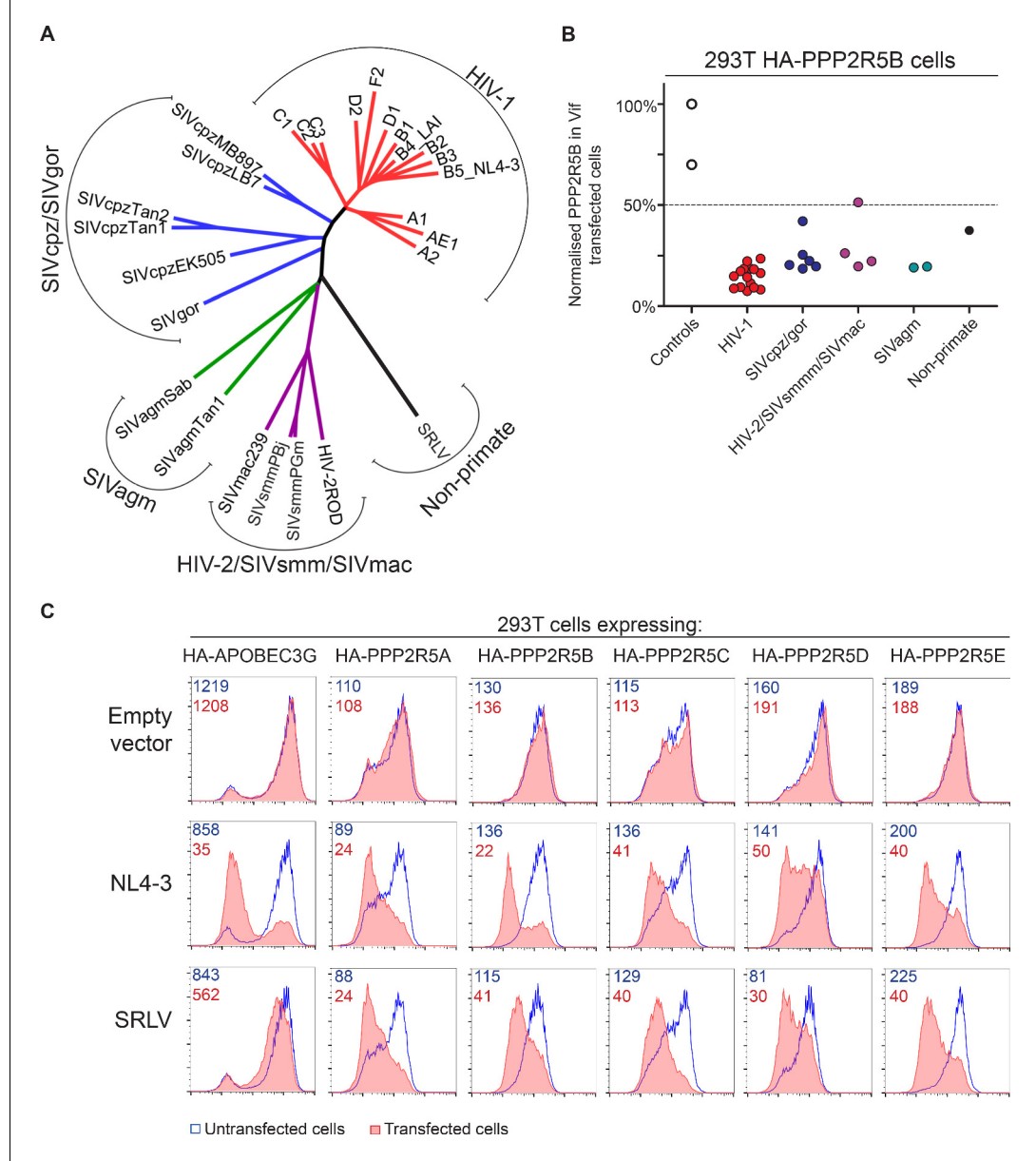

**Figure 7.** Phylogenetic conservation of PPP2R5A-E subunit degradation. (**A**) Phylogenetic tree based on the amino acid alignment of Vif variants used in this study. (**B**) Conservation of PPP2R5B subunit degradation by phylogenetically diverse lentiviral Vif variants. 293T cells stably expressing HA-PPP2R5B were transfected with a panel of lentiviral Vif variants and analysed by intracellular flow cytometry for HA 36 hr post-transfection. The median fluorescence intensity of the transfected population is shown as a proportion of median fluorescence intensity of the untransfected population for each condition, normalized to the empty vector control. Datapoints represent mean values for different Vif variants obtained from up to four independent experiments. (**C**) Depletion of PPP2R5A-E subunits by small ruminant lentivirus Vif. 293T cells stably expressing HA-PPP2R5A-E or HA-APOBEC3G were transfected with NL4-3 (HIV-1) or SRLV Vif variants. Histograms show GFP positive (transfected, red shading) and negative (untransfected, blue line) cells. MFI values are shown for GFP positive (red) and negative (blue) cells.

The following figure supplements are available for figure 7:

**Figure supplement 1.** Identity/similarity matrix of lentiviral Vif variants.

**Figure supplement 2.** Additional data on phylogenetic conservation of PPP2R5A-E subunit degradation.

**Figure supplement 3.** Mechanism of PPP2R5E degradation by SRLV Vif.

Membrane Profiling; PMP) to quantitate 804 plasma membrane proteins (*Matheson et al., 2015*). Although 1030 proteins quantitated in our whole cell proteomic analysis also had Gene Ontology Cellular Component annotations suggesting localisation to the plasma membrane, there was limited overlap with our PMP dataset (*Figure 1—figure supplement 1H*, **upper panel**). The techniques are therefore non-redundant, and this is likely to reflect differential enrichment of scarce or poorly soluble/aggregate-prone glycoproteins using PMP, and intrinsic transmembrane proteins lacking significant extracellular domains or glycosylation sites using whole cell proteomics (*Figure 1—figure supplement 1H*, **lower panel**).

In our earlier temporal proteomic study of HCMV infection, we utilised temporal classification of cellular protein expression to predict novel immunoreceptors (*Weekes et al., 2014*). Here, we develop and extend this methodology to predict cellular targets of specific HIV proteins. Whereas HCMV encodes >150 canonical ORFs (*Wilkinson et al., 2015*), HIV-1 encodes only 9 genes and 15 proteins. Amongst these, the accessory proteins Vif, Vpr, Vpu and Nef have distinct patterns of temporal expression, and HIV-1 is therefore ideally suited to this approach. Vpu is translated from the same transcripts as Env (*Schwartz et al., 1990*), and therefore expressed late in the viral replication cycle (*Figure 1B*). Accordingly, cell surface proteins targeted specifically by Vpu (tetherin and SNAT1) are depleted late in the time course of infection (*Figure 1—figure supplement 1A*), and intracellular proteins known to be targeted by Vif (APOBEC3C) and Vpr (UNG) show distinct temporal profiles (*Figure 1D*).

Based on the similarity with the temporal profile of APOBEC3C, we predicted that other proteins in cluster #1 (*Figure 3*) would be candidate Vif targets. We validated this prediction by comparing changes in protein expression during WT and ΔVif virus infections, and demonstrated that as with APOBEC3C degradation, Vif was necessary for depletion of the PP2A regulatory subunits PPP2R5A-E. Conversely, like known Vpr-target UNG, proteins in cluster #2 are downregulated early in viral infection, in the absence of Vif, and in the presence of reverse transcriptase inhibitors. Vpr is reported to antagonize DNA repair pathways and inhibit innate immune sensing of viral nucleic acids (*Laguette et al., 2014*; *Schrofelbauer et al., 2005*). Interestingly, cluster #2 is markedly enriched for nucleic acid binding proteins, including proteins from families with known roles in DNA damage repair and nucleic acid sensing (*Figure 2—figure supplement 1B* and *Figure 2—source data 1*). Whilst this manuscript was in preparation, downregulation of a second protein in cluster #2, helicase-like transcription factor (HLTF), was also attributed to Vpr in incoming viral particles (*Hrecka et al., 2016*; *Lahouassa et al., 2016*). Remarkably, as for Vif targets APOBEC3C and PPP2R5A-E, temporal profiles of Vpr targets UNG and HLTF cluster very tightly (*Figure 2—figure supplement 1C–D*). Other proteins in cluster #2 with similar temporal profiles are therefore very strong candidates for novel Vpr targets, and downregulation of these proteins by Vpr may antagonize DNA repair pathways or inhibit viral nucleic acid sensing.

Reversible serine/threonine phosphorylation is the most commonly observed post-translational modification (*Khoury et al., 2011*), and PP1 and PP2A, are the major cellular serine/threonine phosphatases. The core PP2A enzyme consists of one catalytic subunit encoded by *PPP2CA* or *PPP2CB* and one structural subunit encoded by *PPP2R1A* or *PPP2R1B*. The specificity of the holoenzyme is determined by the binding of an additional regulatory subunit, encoded by a total of 15 genes, split into four families (*Yang and Phiel, 2010*). We found Vif-dependent proteasomal degradation of all five members of the B56 family (PPP2R5A-E; also known as the B', PR61 or PPP2R5 family). Since each PP2A holoenzyme contains a single regulatory subunit, it is unlikely that depletion of individual regulatory subunits destabilizes other B56 family members. Given the high sequence similarity between B56 subunits, and the ability of Vif to deplete individual subunits expressed non-stoichiometrically in 293T cells, it is much more likely that degradation is mediated by a conserved Vif interaction domain in all five family members.

PP2A makes up 0.2–1% of total eukaryotic cellular protein (*Lin et al., 1998*; *Ruediger et al., 1991*), and whilst in many cases the relevant regulatory subunits have not been characterized, published targets of PP2A-B56 holoenzymes are nonetheless implicated in a multitude of cellular processes (*Yang and Phiel, 2010*). In order to confirm functional PP2A-B56 antagonism and identify relevant PP2A-B56 substrates in HIV-infected cells, we therefore carried out a comprehensive, unbiased analysis of cellular protein phosphorylation during productive HIV infection, and provide a multiplex TMT-based replicated analysis of >8500 cellular phosphopeptides. As expected, we found enhanced phosphorylation of proteins associated with Vpr-mediated activation of the DNA damage

response and G2/M cell cycle arrest (*Figure 6—figure supplement 1B*), reflecting increased activity of the mammalian checkpoint kinases ATR/ATM (*Figure 6—figure supplement 1C*) (*Lai et al., 2005*; *Nakai-Murakami et al., 2007*; *Roshal et al., 2003*; *Vassena et al., 2013*). Conversely, PP2A-B56 antagonism by Vif resulted in hyperphosphorylation of a more limited subset of host phospho-proteins, mirroring previously reported changes seen with PP2A inhibition using okadaic acid.

Our unbiased analysis of Vif-dependent kinase pathways in HIV-infected cells identified a striking increase in the activity of the aurora kinases (*Figure 6C* and *Figure 6—figure supplement 1C*). Aurora kinase activity and abundance peak in late G2 and mitosis (*Bischoff et al., 1998*; *Ly et al., 2014*), and PP2A-B56 holoenzymes antagonize aurora kinase functions in other systems (*Bastos et al., 2014*; *Espert et al., 2014*; *Kruse et al., 2013*; *Xu et al., 2013*). Conversely, aurora kinase activity is typically inhibited by the DNA damage response (*Bensimon et al., 2011*), and reduced activity would therefore be expected in HIV-infected cells. Instead, we propose that Vif-mediated antagonism of PP2A-B56 sustains aurora kinase activity in the presence of the DNA damage response. Interestingly, PLK1 is also inhibited by the DNA damage response in other systems (*Bensimon et al., 2011*), but kinase-active PLK1 is recruited to the SLX4 complex by Vpr in HIV-infected cells (*Laguette et al., 2014*), consistent with the results of this study (*Figure 6E*). Manipulation of mitotic kinases is therefore a shared feature of the HIV accessory proteins Vpr and Vif, and pharmacological inhibitors targeting these cellular kinases may represent a viable antiviral strategy.

Replication of WT and ΔVif viruses in vitro is equivalent in permissive cell lines lacking APOBEC3G expression (*Sheehy et al., 2002*). Aurora kinase activity controls Lck kinase location and phosphory-lation at the immunological synapse (*Blas-Rus et al., 2016*), and kinase-active Lck is also recruited to the virological synapse during cell-cell transmission of HIV (*Vasiliver-Shamis et al., 2009*). It is there-fore possible that Vif-mediated PP2A-B56 antagonism directly enhances cell-cell spread in vivo or in vitro in primary T cells or macrophages, but it is not currently practicable to compare replication of WT and ΔVif viruses on an APOBEC family-negative background in primary cells. Alternatively, PP2A-B56 antagonism may enhance HIV replication or persistence in vivo indirectly, by modulating T cell activation or macrophage polarization (*Blas-Rus et al., 2016*; *Ding et al., 2015*). It is also possi-ble that in other cell types or systems, such as terminally differentiated (non-cycling) macrophages, signalling through alternative kinases may be differentially amplified by Vif-mediated PP2A-B56 depletion. Nonetheless, since PP2A-B56 antagonism spans lineages of lentiviruses which are primar-ily tropic for both lymphocytes (primate lentiviruses) and myeloid cells (non-primate lentiviruses), it is likely that modulation of key kinases is conserved across cell types.

The significance of host factors targeted by HIV is proven in vivo by evolutionary conservation of antagonism across a range of HIV and SIV viruses, and by the existence of similar mechanisms in other viruses. For example, MHC class I proteins are targeted by Nef variants of all primate lentivi-ruses (*Specht et al., 2008*), and manipulation of MHC class I is a common attribute of many virus families (*Randow and Lehner, 2009*). Here, we show that the degradation of PP2A-B56 subunits is conserved across Vif variants from diverse HIV and SIV lentiviruses of primates, as well as a small ruminant lentivirus of sheep (SRLV). The lentiviral genus is ancient (*Gifford et al., 2008*; *Katzourakis et al., 2007*; *Keckesova et al., 2009*; *Worobey et al., 2010*), and species-specific line-ages have developed through virus-host co-evolution. Accordingly, the most recent common ances-tor of the primate and small ruminant lentiviruses is likely to have existed in the common ancestor of primates and ruminants, approximately 100 million years ago (*Hedges et al., 2015*). We therefore propose that degradation of PP2A-B56 subunits is a primordial feature of Vif, present in the com-mon ancestor of primate lentiviral and SRLV Vif variants. Alternatively, Vif variants from these line-ages may have independently acquired this ability. Either possibility strongly suggests a critical selective advantage for lentiviral replication or persistence in vivo.

## Materials and methods

### General cell culture

CEM-T4 T cells (AIDS Reagent Program, Division of AIDS, NIAD, NIH: Dr. JP Jacobs) (*Foley et al., 1965*) were cultured in RPMI supplemented with 10% FCS, 100 units/ml penicillin and 0.1 mg/ml streptomycin at 37°C in 5% CO2. HEK 293T cells and HeLa cells (Lehner laboratory stocks) (*Matheson et al., 2015*) were cultured in DMEM supplemented with 10% FCS, 100 units/ml

penicillin and 0.1 mg/ml streptomycin at 37°C in 5% CO2. All cells were confirmed to be myco-plasma negative (MycoAlert, Lonza, Switzerland). Cell line authentication was not undertaken.

## Stable isotope labelling with amino acids in cell culture (SILAC)

For SILAC labelling, CEM-T4 T cells were grown for at least 7 cell divisions in SILAC RPMI lacking lysine and arginine (Thermo Scientific, Thermo Fisher Scientific, UK) supplemented with 10% dialysed FCS (Gibco, Thermo Fisher Scientific), 100 units/ml penicillin and 0.1 mg/ml streptomycin, 280 mg/L proline (Sigma, UK) and light (K0, R0; Sigma), medium (K4, R6; Cambridge Isotope Laboratories, Tewksbury, MA) or heavy (K8, R10; Cambridge Isotope Laboratories) $^{13}$C/$^{15}$N-contain-ing lysine (K) and arginine (R) at 50 mg/L.

## Primary cell isolation and culture

Primary human CD4$^+$ T cells were isolated from peripheral blood by density gradient centrifugation over Lympholyte-H (Cedarlane Laboratories, Canada) and negative selection using the Dynabeads Untouched Human CD4 T Cells kit (Invitrogen, Thermo Fisher Scientific) according to the manufac-turer's instructions. Purity was assessed by flow cytometry for CD3 and CD4 and routinely found to be ≥95%. Cells were activated using Dynabeads Human T-Activator CD3/CD28 beads (Invitrogen) according to the manufacturer's instructions and cultured in RPMI supplemented with 10% FCS, 30 U/ml recombinant human IL-2 (PeproTech, Rocky Hill, NJ), 100 units/ml penicillin and 0.1 mg/ml streptomycin at 37°C in 5% CO$_2$.

## HIV molecular clones

pNL4-3-dE-EGFP (derived from the HIV-1 molecular clone pNL4-3 but encoding Enhanced Green Fluorescent Protein (EGFP) in the *env* open reading frame (ORF), rendering Env non-functional) was obtained through the AIDS Reagent Program, Division of AIDS, NIAD, NIH: Drs Haili Zhang, Yan Zhou, and Robert Siliciano (*Zhang et al., 2004*) and the complete sequence verified by Sanger sequencing (Source BioScience, UK).

To generate a Vif-deficient clone, overlapping PCR mutagenesis was used to introduce a stop codon early in the Vif ORF, after the final in-frame start codon, as shown below.

## Wild type sequence from start of Vif ORF (bold indicates in-frame start codons)

**ATG**GAAAACAGATGGCAGGTG**ATG**ATTGTGTGGCAAGTAGACAGG**ATG**AGGATTAACACA
TGGAAAAGATTAGTAAAACACCAT**ATG**TATATT

## Mutagenised sequence (underlined region indicates introduced stop codons)

**ATG**GAAAACAGATGGCAGGTG**ATG**ATTGTGTGGCAAGTAGACAGG**ATG**AGGATTAACACA
TGGAAAAGATTAGTAAAACACCAT**ATG**<u>TAATAA</u>

Restriction fragments were subcloned back into pNL4-3-dE-EGFP and mutations verified by Sanger sequencing (Source BioScience) and immunoblot of infected CEM-T4s for Vif protein.

## Vectors for transgene expression

For lentiviral transgene expression in 293T cells, N-terminal 4xHA tagged PP2R5 genes were subcl-oned from pCEP-4xHA-PPP2R5A-E (a kind gift from Dr. David Virshup, Addgene plasmids #14532–14537 [*Seeling et al., 1999*]) into pHRSIN-PGK-puro (*van den Boomen et al., 2014*). APOBEC3G-HA was subcloned from pcDNA3.1-APOBEC3G-HA (AIDS Reagent Program, Division of AIDS, NIAD, NIH: Dr. Warner C Greene [*Sheehy et al., 2002*; *Stopak et al., 2003*]).

PcVif, PcΔVif, and PcVif C114S expression vectors were a kind gift from Prof. Michael Malim, and have been previously described (*Huthoff and Malim, 2007*). pCRV1 Vif expression vectors for HIV-1 and primate lentiviral Vif variants were a kind gift from Prof. Viviana Simon and have been previously described (*Binka et al., 2012*; *Letko et al., 2013*). SRLV Vif was subcloned into pCRV1 by PCR from an SRLV EV1 Vif expression cloning vector, a kind gift from Dr. Barbara Blacklaws, University of Cam-bridge (*Wu et al., 2008*). The dominant negative (DN) CUL5 expression plasmid pcDNA3-DN-hCUL5-FLAG was a kind gift from Prof. Wade Harper (Addgene plasmid #15823 [*Jin et al., 2005*]).

## Lentivectors for shRNA expression

For lentiviral shRNA-mediated knockdown of EloB (*TCEB2*), EloC (*TCEB1)* and CBFβ (*CBFB*) in 293T cells, hairpins were cloned into pHRSIREN-PGK-hygro (related to pHRSIREN-PGK-SBP-ΔLNGFR-W, but expressing hygromycin resistance (*Matheson et al., 2014*).The following oligonucleotides were inserted using BamHI-EcoRI (only top oligonucleotides are shown). Gene specific target sequences are underlined and the source of the target sequence design shown in parentheses.

EloB (Broad Institute GPP web portal at https://www.broadinstitute.org/)
GATCCGCCACAAGACCACCATCTTTATTCAAGAGATAAAGATGGTGGTCTTGTGGCTTTTTTG

EloC (Takara Clontech RNAi design tool at http://bioinfo.clontech.com/rnaidesigner)
GATCCGCACCGAGATTCCTGAATTCTTCAAGAGAGAATTCAGGAATCTCGGTGTTTTTTACGCGTG

Cbfβ (Broad Institute GPP web portal at https://www.broadinstitute.org/)
GATCCGAAGATAGAGACAGGTCTCATTTCAAGAGAATGAGACCTGTCTCTATCTTCTTTTTTG

## Viral stocks

VSVg-pseudotyped NL4-3-dE-EGFP HIV viral stocks were generated by co-transfection of 293T cells with pNL4-3-dE-EGFP molecular clones and pMD.G at a ratio of 9:1 (μg) DNA and a DNA:FuGENE 6 (Promega, UK) ratio of 1 μg:6 μl. Media was changed the next day and viral supernatants harvested and filtered (0.45 μm) at 48 hr prior to concentration with LentiX Concentrator (Clontech, Takara Bio Europe, France) and storage at −80°C. VSVg-pseudotyped pHRSIN and pHRSIREN lentivector stocks were generated by co-transfection of 293T cells with lentivector, p8.91 and pMD.G at a ratio of 2:1:1 (μg) DNA and a DNA:FuGENE 6 ratio of 1 μg:3 μl. Viral supernatants were harvested, filtered, concentrated if required and stored at −80°C. NL4-3-dE-EGFP HIV viral stocks were titred by infection/transduction of known numbers of relevant target cells under standard experimental conditions followed by flow cytometry for GFP and CD4 at 48–72 hr to identify % infected cells.

## CEM-T4 T cell infections

CEM-T4 T cells were infected with concentrated HIV viral stocks by spinoculation at 800 ×g for 2 hr in a non-refrigerated benchtop centrifuge in complete media supplemented with 10 mM HEPES. In experiments with reverse transcriptase inhibitors, cells were incubated with zidovudine (10 μM) and efavirenz (100 nM) (AIDS Reagent Program, Division of AIDS, NIAD, NIH) for 1 hr prior to spinoculation, and inhibitors maintained at these concentrations during subsequent cell culture.

## Tandem mass tag (TMT)-based whole cell proteomic time course analysis

### Sample preparation

For the TMT-based HIV infection time course, CEM-T4 T cells were spinoculated with VSVg-pseudotyped NL4-3-dE-EGFP HIV at a multiplicity of infection (MOI) of 10 in the presence or absence of reverse transcriptase inhibitors. Aliquots of cells were harvested sequentially at the indicated timepoints, and dead cells removed using the Miltenyi Dead Cell Removal kit (Miltenyi Biotec, Germany). Cells were analysed by flow cytometry for CD4 and GFP expression (confirming 95% productive infection) and subjected to Plasma Membrane Profiling (PMP; plasma membrane proteomic analysis) and this data has been previously published (*Matheson et al., 2015*). For whole cell proteomic analysis, $2 \times 10^6$ viable cells per timepoint were then washed with ice-cold PBS with Ca/Mg pH 7.4 (Sigma) and frozen at −80°C prior to Filter Aided Sample Preparation (FASP) essentially as previously described (*Wisniewski et al., 2009*)

Cell pellets were thawed on ice, lysed in 4% SDS/100 mM HEPES (Sigma) supplemented with Complete Protease Inhibitor Cocktail (without EDTA; Roche, UK), and sonicated at 4°C using a Diagenode Bioruptor. Protein concentrations were determined using the Pierce BCA Protein Assay kit (Thermo Scientific) and 100 μg protein per timepoint subjected to downstream processing. Lysates

were transferred to Microcon-30 kDa Centrifugal Filter Units, reduced (100 mM DTT) and alkylated (50 mM iodoacetamide) at room temperature, washed with a total of 5 column volumes of 8 M urea/100 mM HEPES and 100 mM HEPES pH 8.5, then incubated with 1 µg/50 µl modified sequencing grade trypsin (Promega) in 100 mM HEPES pH 8.5 at 37°C for 8 hr. After digestion, peptide eluates were collected by centrifugation and stored at +4°C prior to TMT labelling the next day.

For TMT labelling, TMT 6-plex reagents (Thermo Scientific) were dissolved in anhydrous acetonitrile (0.8 mg/40 µl) according to the manufacturer's instructions. Peptide concentrations were determined using the Pierce Micro BCA Protein Assay kit (Thermo Scientific) and 25 µg peptide per sample labelled with 20 µl reconstituted TMT 6-plex reagent at a final acetonitrile concentration of 30% (v/v). Samples were labelled as follows: uninfected cells, 0 hr (TMT 126); 6 hr (TMT 127); 24 hr (TMT 128); 48 hr (TMT 129); 72 hr (TMT 130); 72 hr plus reverse transcriptase inhibitors (TMT 130). Following incubation at room temperature for 1 hr, reactions were quenched with hydroxylamine to a final concentration of 0.3% (v/v). Samples were mixed at a ratio of 1:1:1:1:1:1 and dried down to remove acetonitrile prior to off-line peptide fractionation.

## Off-line high pH reversed-phase (HpRP) peptide fractionation

TMT-labelled tryptic peptides were subjected to HpRP-HPLC fractionation using a Dionex Ultimate 3000 powered by an ICS-3000 SP pump with an Agilent ZORBAX Extend-C18 column (4.6 mm × 250 mm, 5 µm particle size). Mobile phases (H$_2$0, 0.1% NH$_4$·H or MeCN, 0.1% NH$_4$·H) were adjusted to pH 10.5 with the addition of formic acid and peptides were resolved using a linear 40 min 0.1–40 % MeCN gradient over 40 min at a 400 µl/min flow rate and a column temperature of 15°C. Eluting peptides were collected in 15 s fractions. One hundred and twenty fractions covering the peptide-rich region were re-combined to give 10 samples for analysis. To preserve orthogonality, fractions were combined across the gradient, with each of the concatenated samples comprising 12 fractions which were 10 fractions apart. Re-combined fractions were dried down using an Eppendorf Concentrator (Eppendorf, UK) and resuspended in 15 µl MS solvent (3% MeCN, 0.1% TFA)

## Mass spectrometry

Data for TMT labelled samples were generated using an Orbitrap Fusion Tribrid mass spectrometer (Thermo Scientific). Peptides were fractionated using an RSLCnano 3000 (Thermo Scientific) with solvent A comprising 0.1% formic acid and solvent B comprising 80% MeCN, 20% H$_2$O, 0.1% formic acid. Peptides were loaded onto a 50 cm Acclaim PepMap C18 column (Thermo Scientific) and eluted using a gradient rising from 10 to 25% solvent B by 90 min and 40% solvent B by 115 min at a flow rate of 250 nl/min. MS data were acquired in the Orbitrap at 120,000 fwhm between 400–1600 m/z. Spectra were acquired in profile with AGC 4 × 10$^5$. Ions with a charge state between 2+ and 6+ were isolated for fragmentation in top speed mode using the quadrupole with a 1.5 m/z isolation window. CID fragmentation was performed at 35% collision energy with fragments detected in the ion trap between 400–1200 m/z. AGC was set to 5 × 10$^3$ and MS2 spectra were acquired in centroid mode. TMT reporter ions were isolated for quantitation in MS3 using synchronous precursor selection. Ten fragment ions were selected for MS3 using HCD at 53% collision energy. Fragments were scanned in the Orbitrap at 60,000 fwhm between 100–500 m/z with AGC set to 2 × 10$^5$. MS3 spectra were acquired in profile mode with injection parallelisation enabled.

## Data processing and analysis

Raw MS files were processed using Proteome Discoverer 1.4.0.288 (Thermo Scientific). Data were searched against a concatenated human (UniProt, downloaded on 04/11/13) and HIV-1 (based on pNL4-3, Genbank: AF324493.2) database as previously described (*Matheson et al., 2015*). VSVg (UniProt: P03522) and dEnv-EGFP-KDEL (inferred from the pNL4-3-dE-EGFP sequence) were substituted for Env. Precursor mass tolerance and fragment mass tolerance were set to 10 ppm and 0.6 Da, respectively, with a maximum of two missed tryptic cleavage sites. Percolator was used for post-processing of search results with a peptide false discovery rate of 0.01. Observed reporter ion intensities were adjusted for lot-specific isotopic impurities and missing quan values replaced with the minimum detected ion intensity. Protein abundances were calculated using unique peptides and normalised according to median protein ratios.

The complete HIV-1 infection time course mass spectrometry proteomics dataset has been deposited to the ProteomeXchange Consortium (*Vizcaino et al., 2013*) via the PRIDE Proteomics Identifications (*Vizcaino et al., 2013*) partner repository with the dataset identifier PXD004187 (accessible at http://proteomecentral.proteomexchange.org).

For Gene Set Enrichment Analysis (GSEA), all proteins identified by >1 unique peptide were analysed using GSEA v2.2.2 (downloaded from http://software.broadinstitute.org/gsea/index.jsp) and KEGG Pathway (c2.cp.kegg.v5.1.symbols.gmt) and Gene Ontology Biological Process (c5.bp.v5.1.symbols.gmt) gene sets from the Molecular Signatures Database (MSigDB) v5.1 (*Mootha et al., 2003*; *Subramanian et al., 2005*). Pairwise comparisons were conducted between uninfected cells and infected cells at each timepoint. Genes were ranked by $log_2$_Ratio_of_Classes and FDR q values were calculated using 1000 gene_set permutations. For charting, nominal FDR q values of 0 were replaced with minimum values for each pairwise comparison.

For clustering according to profiles of temporal expression, proteins identified by >1 unique peptide and with a minimum fold change from baseline (0 hr) of >2 were analysed using Cluster 3.0 (downloaded from http://bonsai.hgc.jp/~mdehoon/software/cluster/software.htm) (*de Hoon et al., 2004*; *Eisen et al., 1998*) and visualised using Java TreeView 1.1.6r4 (downloaded from http://jtreeview.sourceforge.net) (*Saldanha, 2004*). Data were expressed as $log_2$ (fold change in protein abundance compared with uninfected cells) and agglomerative hierarchical clustering performed using uncentered Pearson correlation and centroid linkage (*Eisen et al., 1998*; *Weekes et al., 2014*).

For functional analysis of proteins in clusters #1–4, enrichment of Gene Ontology Biological Process and Molecular Function terms against a background of all proteins quantitated was determined using the Database for Annotation, Visualization and Integrated Discovery (DAVID) 6.7 (accessed on 29/4/16 at http://david.abcc.ncifcrf.gov) with default settings (*Huang da et al., 2009a*, ). Annotation clusters with enrichment scores > 1.3 (equivalent to a geometric means of all included enrichment p values<0.05) were considered significant.

For the interactive spreadsheet of all TMT data, gene name aliases were added using GeneALaCart (accessed on 4/5/16 at https://genealacart.genecards.org) (*Rebhan et al., 1997*).

For comparison with PMP, Gene Ontology Cellular Component (GOCC) terms were imported using Perseus 1.4.1.3 (downloaded from http://maxquant.org). The number of plasma membrane proteins quantitated was estimated by counting proteins with GOCC terms 'plasma membrane', 'cell surface' and 'extracellular' or with short, membrane-specific GOCC terms but no subcellular assignment (*Matheson et al., 2015*). Glycosylation sites were identified from the UniProt Knowledgebase (accessed on 4/12/15 at http://www.uniprot.org). Experimentally identified N- and O-linked glycosylation sites and predicted sites of N- and mucin-type O-linked glycosylation (using the NegNGlyc and NetOGlyc tools) were included.

## Stable isotope labelling with amino acids in cell culture (SILAC)-based proteomic validation time course

### Sample preparation

For the SILAC-based validation time course, CEM-T4 T cells pre-labelled with heavy lysine and arginine were spinoculated with VSVg-pseudotyped NL4-3-dE-EGFP HIV at an MOI of 10, and cells pre-labelled with medium lysine and arginine were mock-spinoculated without virus. Aliquots of HIV-1-infected (heavy) and mock (medium) cells were harvested sequentially at the indicated timepoints, dead cells removed using the Miltenyi Dead Cell Removal kit, and equal cell numbers mixed prior to whole cell proteomic analysis. Cell lysis, protein extraction and digestion and off-line peptide fractionation were carried out essentially as for TMT-based whole cell proteomics, except that 100 mM Tris/HCl pH 7.4 was substituted for 100 mM HEPES in lysis and wash buffers cells, 50 mM ammonium bicarbonate was substituted for 100 mM HEPES in digest buffer, and peptide eluates were not subjected to TMT labelling. Cells were also subjected to PMP and this data has been previously published (*Matheson et al., 2015*).

### Mass spectrometry

Data for SILAC labelled samples were generated using a Q Exactive Orbitrap mass spectrometer (Thermo Scientific). Peptides were fractionated using an RSLCnano 3000 (Thermo Scientific) with solvent A comprising 0.1% formic acid and solvent B comprising 80% MeCN, 20% $H_2O$, 0.1% formic

acid. Peptides were loaded onto a 50 cm C18 EASYspray column (Thermo Scientific) and eluted using a gradient rising from 10% to 36% B by 75 min and 55% B by 100 min. MS data were acquired at 70,000 fwhm between 400–1650 m/z with AGC of $1 \times 10^6$ and 250 ms injection time. MS2 data was acquired at 17,500 fwhm with AGC of $5 \times 10^4$, 200 ms injection time and a loop count of 10. HCD fragmentation was performed at NCE of 28% and an underfill ratio of 20%.

## Data processing and analysis

Raw MS files were processed using MaxQuant 1.3.0.5. Data were searched against a concatenated human (UniProt, downloaded 04/11/13) and HIV-1 (based on pNL4-3, Genbank: AF324493.2) database as previously described (*Matheson et al., 2015*). VSVg (UniProt: P03522) and dEnv-EGFP-KDEL (inferred from the pNL4-3-dE-EGFP sequence) were substituted for Env. Fragment ion tolerance was set to 0.5 Da with a maximum of two missed tryptic cleavage sites. Carbamidomethyl (C) was defined as a fixed modification, oxidation (M), acetylation (protein N-terminal) and deamidation (NQ) were selected as variable modifications. A reversed decoy database was used with the false discovery rate for both peptides and proteins set at 0.01. Peptide re-quantify was enabled and quantitation utilized razor and unique peptides. Normalized protein ratios are reported.

For validation of downregulation or upregulation of proteins in clusters #1–4 at the indicated timepoints, mean $\log_2$ (H/M protein abundance) for proteins in each cluster were compared with 0 (no regulation) using 2-tailed 1-sample T-tests conducted using XLSTAT.

## Tandem mass tag (TMT)-based whole cell proteomic/phosphoproteomic single timepoint analysis

### Sample preparation

For the TMT-based single timepoint analyses, CEM-T4 T cells were mock-spinoculated or spinoculated in triplicate with VSVg-pseudotyped NL4-3-dE-EGFP wildtype and Vif-deficient HIV at an MOI of 1.5. Cells were harvested 48 hr after infection, and dead cells removed using the Miltenyi Dead Cell Removal kit. $2 \times 10^6$ viable cells per condition were washed with ice-cold PBS with Ca/Mg pH 7.4 (Sigma), lysed in 8 M urea, 50 mM TEAB (triethylammonium bicarbonate) pH 8.5 including phosphatase inhibitors (phosSTOP, Roche) and subjected to 10 rounds of sonication (30 s on/off) in a Diagenode Bioruptor sonicator at 4°C. Lysate protein concentrations were quantified using the Pierce BCA Protein Assay kit (Thermo Scientific). 800 μg lysate/replicate was reduced with 10 mM TCEP for 20 min at room temperature and alkylated with 20 mM iodoacetamide (IAM) for 20 min at room temperature in the dark before quenching excess IAM with 15 mM DTT. Digestion was performed by first adding LysC at a 1:100 enzyme:protein ratio and incubating at 30°C for 3 hr. This digest was then diluted 4x with 50 mM TEAB and trypsin was added at a 1:50 enzyme:protein ratio and digested overnight at 37°C with shaking on a Themomixer (Eppendorf). Digests were subsequently acidified with TFA and cleaned up by SPE using SepPak C18 cartridges (Waters, UK). SPE eluates were divided into 50 μg and 750 μg-equivalent aliquots before drying in a vacuum centrifuge. The 50 μg aliquots were resuspended in 100 mM TEAB prior to TMT labelling essentially as per the manufacturer's instructions. TMT-labelled samples were then pooled and dried under vacuum prior to HpRP fractionation, and the sample pool used for whole cell proteome analysis. The 750 μg aliquots were subjected to phosphopeptide enrichment.

### Phosphopeptide enrichment

750 μg-equivalent aliquots were resuspended in a loading solution of final concentration 4% TFA, 1 M glycolic acid and 80% acetonitrile, added to 4.5 mg of 10 μm titanium dioxide resin (Titansphere, GL Sciences, Japan) and shaken vigorously for 30 min. Beads were then washed for 5 min with vigorous shaking with 100 μL of the following solutions: loading solution, 1% TFA 80% ACN and 0.1% TFA 10% ACN. Enriched peptides were eluted with 100 μL 5% ammonium hydroxide for 20 min with vigorous shaking before being acidified with TFA and formic acid. Acidified peptide pools were cleaned up by SPE using SepPak C18 cartridges (Waters) before TMT labelling and HpRP fractionation.

## Off-line high pH reversed-phase (HpRP) peptide fractionation

HpRP fractionation was conducted on an Ultimate 3000 UHPLC system (Thermo Scientific) equipped with a 2.1 mm × 25 cm, 1.7 μ, Kinetex-Evo C18 column (Phenomenex, UK). Solvent A was 3% ACN, Solvent B was 100% ACN, solvent C was 200 mM ammonium formate (pH 10). Throughout the analysis solvent C was kept at a constant 10%. The flow rate was 400 μL/min and UV was monitored at 280 nm. Samples were loaded in 90% A for 10 min before a gradient elution of 0–50% B over 43 min followed by a 10 min wash with 90% B. 15 s (100 μL) fractions were collected throughout the run. Peptide containing fractions were orthogonally recombined into 24 fractions (i.e. fractions 1, 25, 49, 73, 97 combined) and dried in a vacuum centrifuge. Fractions were stored at −80°C prior to analysis.

## Mass spectrometry

Data were acquired on an Orbitrap Fusion mass spectrometer (Thermo Scientific) coupled to an Ultimate 3000 RSLC nano UHPLC (Thermo Scientific). HpRP fractions were resuspended 20 μl 5% DMSO 0.5% TFA. Samples were analysed using a nanoLC-MS platform consisting of an Ultimate 3000 RSLC nano UHPLC coupled to an Orbitrap Fusion instrument (Thermo Scientific). 50% of whole cell and 80% of phosphoproteome fractions were loaded at 10 μl/min for 5 min on to an Acclaim PepMap C18 cartridge trap column (300 um × 5 mm, 5 um particle size) in 0.1% TFA. After loading a linear gradient of 3–32% solvent B over 2 hr was used for sample separation over a column of the same stationary phase (75 μm × 50 cm, 2 μm particle size) before washing at 90% B and re-equilibration. Solvents were A: 0.1% FA and B:ACN/0.1% FA. For whole cell proteome samples electrospray ionisation was achieved by applying 2.1 kV directly to a stainless steel emitter (Thermo Scientific). For phosphopeptide samples a distal coated silica emitter was used (New Objective, Woburn, MA).

An SPS/MS3 acquisition was used for all samples and was run as follows. MS1: Quadrupole isolation, 120'000 resolution, 5e5 AGC target, 50 ms maximum injection time, ions injected for all parallelisable time. MS2: Quadrupole isolation at an isolation width of m/z 1.6, CID fragmentation (NCE 35) with the ion trap scanning out in rapid mode from m/z 120, 5e3 AGC target, 70 ms maximum injection time (150 ms for phosphopeptides), ions accumulated for all parallelisable time in centroid mode. For phosphopeptides multistage activation was enabled and set to trigger upon neutral loss of 79.9663 Da. MS3: In synchronous precursor selection mode the top 10 MS2 ions were selected for HCD fragmentation (65NCE) and scanned out in the orbitrap at 60'000 resolution with an AGC target of 2e4 and a maximum accumulation time of 148 ms, ions were not accumulated for all parallelisable time. The entire MS/MS/MS cycle had a target time of 2 s. Dynamic exclusion was set to +/−10 ppm for 60 s, MS2 fragmentation was trigged on precursor ions 5e3 counts and above.

## Data processing and analysis

Spectra were searched by Mascot within Proteome Discoverer 2.1 against the UniProt Human database (21/03/16). The database included forward and randomised reversed Human database, the HIV proteome previously mentioned as well as a compendium of common contaminants (GPM). The following search parameters were used. MS1 Tol: 10 ppm, MS2 Tol: 0.6 Da, Fixed mods: Carbamidomethyl (C) and TMT (N-term, K), Var mods: Oxidation (M), Enzyme: Trypsin (/P). Phosphopeptide samples also included variable modification of Phosphorylation (STY). MS3 spectra were used for reporter ion based quantitation with a most confident centroid tolerance of 20 ppm. PSM FDR was calculated using Mascot percolator and was controlled at 0.01% for 'high' confidence PSMs and 0.05% for 'medium' confidence PSMs. Phosphopeptide site confidence was assessed using the ptmRS node (the successor to phosphoRS (*Taus et al., 2011*). Reporter signal to noise (s/n) with a cut-off of 10 was used for quantitation. Normalisation was automated and based on total s/n in each channel. Protein/peptide abundance was calculated and output in terms of 'scaled' values, where the total s/n across all reporter channels is calculated and a normalised contribution of each channel is output. Proteins/peptides satisfying at least a 'medium' FDR confidence were taken forth to statistical analysis in R. This consisted of a moderated T-test (Limma) with Benjamini-Hochberg correction for multiple hypotheses to provide a q value for each comparison (*Schwammle et al., 2013*).

For kinase activity profiling, pairwise comparisons in phosphopeptide abundance between mock and WT or ΔVif HIV-infected cells were conducted using PhosFate (accessed at http://phosfate. com/), including all kinases represented by >1 target phosphosite. For a direct comparison of WT and ΔVif HIV-infected cells, phosphopeptides spanning phosphorylation sites annotated in the

PhosPhositePlus database (accessed at http://www.phosphosite.org/) were identified. Mean $\log_2$ (fold change in phosphopeptide abundance) was calculated for each kinase represented by >4 target phosphosites. Further details of manually curated AURKA, AURKB and PLK1 substrates are shown in *Figure 6—source data 2*.

For functional analysis of proteins hyperphosphorylated in the presence of HIV infection, enrichment of Gene Ontology Biological Process and Molecular Function terms against a background of all identified phosphoproteins was determined using the Database for Annotation, Visualization and Integrated Discovery (DAVID) 6.7 (accessed on 29/4/16 at http://david.abcc.ncifcrf.gov) with default settings (*Huang da et al., 2009a*, *2009b* Proteins containing phosphopeptides significantly upregulated (q values < 0.01) in cells infected with WT HIV-1 compared with mock infected cells were analysed. Annotation clusters with enrichment scores > 1.3 (equivalent to a geometric means of all included enrichment p values<0.05) were considered significant.

## Stable cell lines

Stable 293T cell lines were generated by transduction with pHRSIN-PGK-puro-4xHA-PPP2R5A-E or pHRSIN-PGK-puro-APOBEC3G-HA and selection with puromycin at 1 µg/ml. For knockdown experiments, 293T cells transduced with HA-PPP2R5B or APOEC3G-HA were subsequently transduced with pHRSIREN (control or gene-specific shRNA expression) and selected with hygromycin at 200 µg/ml.

## Flow cytometry

### Regulation of exogenous PPP2R5 subunits by Vif expression in stable 293T cell lines

293T cells stably expressing HA-PPP2R5A-E or APOBEC3G-HA were transfected in 24 well plates using Fugene 6, with 200 ng Vif expression vector and 20 ng pMAXGFP (Lonza). 24 hr later, the medium was changed, and 12 hr subsequently (36 hr post-transfection) cells were harvested for flow cytometry. Briefly, cells were dissociated using PBS/EDTA and fixed and permeabilised using a commercial kit (Cytofix/Cytoperm, BD Biosciences, UK). Permeabilised cells were stained with a fluorescently-conjugated anti-HA antibody, washed, and acquired on a BD FACScalibur or BD LSRFortessa (BD Biosciences). Where indicated, DMSO or bortezomib (10 nM) were added when the media was changed 24 hr post-transfection. For CUL5 WT/DN co-transfection, 100 ng of Vif expression vector was used, with 100 ng of CUL5 WT or DN expression vector. Where 293T cells expressing GFP-tagged PPP2R5B were used, pCMV-SPORT6-mCherry was substituted for pMAXGFP, and cells were analysed without permeabilisation.

### Regulation of endogenous PPP2R5D by HIV infection of T cells

CEM-T4 T cells or activated primary human CD4$^+$ T cells were analysed as described for HA-PPP2R5A-E, but cells were infected with NL4-3-dE-EGFP WT or ΔVif viruses, and stained with unconjugated anti-PPPR5D followed by an AF647-conjugated secondary antibody. For time course analyses in CEM-T4 T cells, bortezomib (20 nM) or cycloheximide (50 µg/ml) were added 24 hr post-infection, and median fluorescence intensity (MFI) values for GFP positive (infected) cells compared at four timepoints (0, 4, 8 and 12 hr). Where indicated, cells were stained for CD4 or ICAM3 without permeabilisation.

## Antibodies

The following primary antibodies were used for immunoblot (alphabetical order): anti-APOBEC3G (AIDS Reagent Program, Division of AIDS, NIAID, NIH from Immunodiagnostics, 10069), anti-ß-catenin (ab32572, Abcam, UK), anti-calreticulin (PA3-900, Thermo Scientific), anti-FLAG (M2, Sigma), anti-HA (16B12, Biolegend), anti-p24 (ab9071, Abcam), anti-PPP2R5A (ab89621, Abcam), anti-PPP2R5D (ab88075, Abcam), anti-PPP2R5D (EPR15617, ab188323, Abcam), anti-RRM2 (ab57653, Abcam), anti-Vif (#319, AIDS Reagent Program, Division of AIDS, NIAID, NIH: Dr. Michael Malim #6459 (*Fouchier et al., 1996*; *Simon et al., 1997*).

The following primary antibodies were used for flow cytometry: anti-CD4-AF647 (clone OKT4; BioLegend, UK), anti-HA-DyLight 650 (16B12, ab117515, Abcam), anti-PPP2R5D (EPR15617, ab188323, Abcam), anti-ICAM3 (TU41, 555957, BD).

The following primary antibodies were used for immunoprecipitation anti-PPP2R5D (EPR15617, ab188323, Abcam).

The following secondary antibodies were used: goat anti-mouse-AF647 and donkey anti-rabbit-AF647 (flow cytometry, Molecular Probes, Thermo Fisher Scientific); goat anti-mouse-HRP and anti-rabbit-HRP (immunoblot, Jackson ImmunoResearch, West Grove, PA).

## Immunoblotting

CEM-T4 T cells or 293T cells were typically lysed in TBS/2% SDS supplemented with Benzonase (Sigma) to reduce lysate viscosity. Lysates were heated in Laemlli Loading Buffer for 15 min at 95°C, separated by SDS-PAGE and transferred to Immobilon-P membrane (Millipore, UK). Membranes were blocked in PBS/5% non-fat dried milk (Marvel)/0.2% Tween and probed with the indicated primary antibody overnight at 4°C. Reactive bands were visualised using HRP-conjugated secondary antibodies and SuperSignal West Pico or Dura chemiluminescent substrates (Thermo Scientific).

## Immunoprecipitation

CEM-T4 T cells or 293T cells were lysed in 1 % NP-40. Lysates were pre-cleared with Protein A-Sepharose (Sigma) or IgG-Sepharose (GE Healthcare, UK) and incubated for 16 hr at 4°C with anti-HA coupled to agarose beads (Sigma EZview Red Anti-HA Affinity Gel) or anti-PPP2R5D/Protein A-Sepharose (Sigma). After washing in 0.5% NP-40, samples were eluted with 0.5 mg/ml HA peptide at 37°C for 1 hr (anti-HA immunoprecipitation) or in Laemlli Loading Buffer without DTT at 70°C for 10 min (anti-PPP2R5D immunoprecipitation). Samples were separated by SDS-PAGE, and immuno-blotted as described.

## Pulse-chase

CEM-T4 T cells were starved for 20 min in methionine-free, cysteine-free RPMI/5% dialysed FCS (Invitrogen), labeled with $[^{35}S]$methionine/$[^{35}S]$cysteine (EasyTag EXPRESS, PerkinElmer) for 15 min, then chased in RPMI/10% FCS at 37°C. Cells were lysed in 1% Triton X-100 at the indicated time-points, and subjected to immunoprecipitation with anti-PPP2R5D as described. Samples were separated by SDS-PAGE and processed for autoradiography using the Packard Cyclone Storage Phosphor System.

## Infectious viral release

pCMV-SPORT6 expression vectors encoding APOBEC3G, tetherin, TFAP4 and FMR1 were obtained from the MGC/IMAGE clone collection (Dharmacon, GE Healthcare) with the following identifiers: APOBEC3G (IMAGE:3905631), BST-2 (IMAGE:5217945), TFAP4 (IMAGE:4181538) and FMR1 (IMAGE:30347992). As a control, mCherry was subcloned into pCMV-SPORT6. 293T cells were transfected in 24 well plates using Fugene 6. Each well received a transfection mix containing 135 ng NL4-3-dE-EGFP and 15 ng pMD.G. Shortly after, each well received a second transfection mix of 150 ng pCMV-SPORT6 mCherry, APOBEC3G, tetherin, TFAP4 or FMR1. The media was changed 24 hr post-infection, and 48 hr post-infection, cell supernatants were harvested. Contaminating 293T cells in the supernatants were removed by centrifugation, and a small proportion used to infect a fixed number of HeLa cells. After 48 hr, the HeLa cells were analysed by flow cytometry to determine the proportion that had become GFP positive (infected). The MOI in each well was calculated and normalized to the MOI resulting from the supernatants of 293T cells receiving mCherry.

## Acknowledgements

The authors thank Dr. Viviana Simon (Icahn School of Medicine at Mount Sinai), Dr. Barbara Black-laws (University of Cambridge), and Prof. Michael Malim (King's College London) for providing reagents, Dr. Jenny Ho (Thermo) for help with proteomics, Dr. Yagnesh Umrania (CIMR-IMS Proteomics Facility) for help with bioinformatics, Dr. Reiner Schulte and the CIMR Flow Cytometry Core Facility team, and the Lehner laboratory for critical discussion. This work was supported by a Wellcome Trust PRF (101835/Z/13/Z) to PJL and RTF to NJM (093964/Z/10/Z), NHSBT and the NIHR Cambridge BRC, a Wellcome Trust Strategic Award to CIMR, and the Addenbrooke's Charitable Trust. NJM is a Raymond and Beverly Sackler student.

## Additional information

### Funding

| Funder | Grant reference number | Author |
|---|---|---|
| Wellcome Trust | 101835/Z/13/Z | Paul J Lehner |
| Wellcome Trust | 093964/Z/10/Z | Nicholas J Matheson |

The funders had no role in study design, data collection and interpretation, or the decision to submit the work for publication

### Author contributions

EJDG, NJM, Conception and design, Acquisition of data, Analysis and interpretation of data, Drafting or revising the article; KW, DJHvdB, RA, Acquisition of data; JCW, Acquisition of data, Analysis and interpretation of data; PJL, Conception and design, Analysis and interpretation of data, Drafting or revising the article

### Author ORCIDs

Nicholas J Matheson, http://orcid.org/0000-0002-3318-1851
Paul J Lehner, http://orcid.org/0000-0001-9383-1054

## Additional files

### Supplementary files

• Supplementary file 1. Previous proteomic studies of HIV-infected cells. Studies cited: *Araínga et al., 2015*; *Haverland et al., 2014*; *Navare et al., 2012*; *Kraft-Terry et al., 2011*; *Rasheed et al., 2009*; *Chan et al., 2007*; *Chan et al., 2009*; *Ringrose et al., 2008*; *Pathak et al., 2009*.

### Major datasets

The following dataset was generated:

| Author(s) | Year | Dataset title | Dataset URL | Database, license, and accessibility information |
|---|---|---|---|---|
| Greenwood EJD, Matheson NJ, Wals K, Antrobus R, Williamson JC, Lehner PJ | 2016 | Temporal proteomic analysis of HIV-infection reveals remodelling of the cellular phosphoproteome by phylogenetically diverse lentiviral Vif variants | http://www.ebi.ac.uk/pride/archive/projects/PXD004187 | Publicly available at the EMBL PRIDE Archive (accession no: PXD004187) |

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
