## [Decision Letter]

Thank you for submitting your article "Temporal proteomic analysis of HIV infection reveals remodelling of the host phosphoproteome by lentiviral Vif variants" for consideration by *eLife*. Your article has been favorably evaluated by Wenhui Li (Senior Editor) and three reviewers, one of whom is a member of our Board of Reviewing Editors. The reviewers have opted to remain anonymous.

The reviewers have discussed the reviews with one another and the Reviewing Editor has drafted this decision to help you prepare a revised submission. While all reviewers found your manuscript of interest, as you can see below, a number of issues were raised.

Summary:

This paper by Greenwood and colleagues applies advanced proteomic methods to address the complex question of what cellular factors are modulated by HIV infection. To this end, the authors examined the abundance of more than 6500 cellular proteins over time in HIV-1 infected T cells using Tandem mass tagging and performed a validation of the data using conventional stable isotope labeling with amino acids in cell culture (SILAC). Systematic analysis of cells infected with HIV-1 that are either Vif proficient or deficient reveals that the B56 family of regulatory proteins (PPPR5 A-E) of the PP2A phosphatase are depleted with kinetics similar to the known Vif target, APOBEC3C. Vif is found sufficient to deplete PPP5A-E subunits and requires all the known components of the Vif/E3 ligase complex, including CUL5, ELOB/C, CBF-β, as well as the proteasome. Through phosphoproteomic analysis of infected cells, the authors confirm that Vif induces the hyperphosphorylation of more than 200 cellular proteins, particularly substrates of the aurora kinases. Importantly, Vif-mediated depletion of PPP2R5A-E emerges as a function conserved across Vif variants from diverse HIV and SIV viruses, suggesting an important role in viral replication or persistence.

Essential revisions:

1) An aspect of the study design has implications for the Resource aspect of this paper. In the 6-plex TMT experiment, the authors do not collect their quantitative data with biological duplicates or triplicates, as is most often done in these types of experiments. As such, there could be uncertainty in the data, particularly for those proteins quantified on the basis of a single peptide (which reflects about 10% of the data based on [Supplementary-material SD1-data]). On the positive side, the authors do a lot of follow-up based on the screen which fits a lot of the known biology and gives confidence in the bulk of conclusions made. This includes a conventional SILAC experiment (Figure 2). Of note, however, there are several proteins in each cluster that do not appear to behave as with the TMT data, suggesting a disconnect for a small number of proteins. Perhaps the use of a single replicate in Figure 1 could be viewed as a screen and as such the manuscript could be considered as a regular article rather than a resource.

2) While the reviewers do appreciate that Figure 4 and Figure 5 firmly establishes PPP2R5 depletion in a 293T co-transfection system requires Cul5, ELOb/C, CBF-β and a functional proteasome, taken as a whole this single model system is not enough to prove that HIV-1 Vif acts specifically on PPP2R5 proteins to induce their degradation in natural infection. Minimally, the authors should provide evidence that HIV-1 Vif does interact with PPP2R5A-E subunits in HIV-infected T cells and forms a complex containing Cul5 components. Furthermore, direct evidence that Vif mediates the degradation of PP2A-B56 subunits in HIV-1 infected T cells should be demonstrated by using pulse-chase experiments or cycloheximide chase assay.

3) Figure 7: The requirement for CBF-β function suggests that the canonical APOBEC-degrading Vif ligase is also responsible for degradation of the PPP2R5 proteins. If this is the case, and based on many prior studies showing that Vif is intrinsically disordered, then this interaction should be primate specific and CBF-β requiring. The non-primate lentiviral Vif proteins clearly function without CBF-β so reviewers strongly question the single MVV Vif data point in Figure 7. We strongly suggest removing these data from the paper and temper the claims of complete conservation. Alternatively, repeat the experiment in Figure 7 with SRLV(MVV) plus/minus Vif to prove that the downregulation of PPP2R5 components is indeed Vif-dependent. The claim of conservation is very important and needs to be carefully investigated and justified.

---

## [Author Response]

*Essential revisions:*

*1) An aspect of the study design has implications for the Resource aspect of this paper. In the 6-plex TMT experiment, the authors do not collect their quantitative data with biological duplicates or triplicates, as is most often done in these types of experiments. As such, there could be uncertainty in the data, particularly for those proteins quantified on the basis of a single peptide (which reflects about 10% of the data based on [Supplementary-material SD1-data]). On the positive side, the authors do a lot of follow-up based on the screen which fits a lot of the known biology and gives confidence in the bulk of conclusions made. This includes a conventional SILAC experiment (Figure 2). Of note, however, there are several proteins in each cluster that do not appear to behave as with the TMT data, suggesting a disconnect for a small number of proteins. Perhaps the use of a single replicate in Figure 1 could be viewed as a screen and as such the manuscript could be considered as a regular article rather than a resource.*

Thank you for your comments – we agree that the distinction between a “resource” and a “research article” is not always straight forward. In this study, we utilised TMT-based multiplexing to compare both different timepoints (Figure 1) and replicates at a single timepoint (Figure 4), and validated our timecourse data using an independent quantitative technique (SILAC) rather than a simple TMT-based biological replicate (Figure 2).

Data from all three experiments are summarised for proteins in clusters #1-4 in [Supplementary-material SD2-data]. Despite the different approaches, the data are remarkably similar, with 139/153 (91%) of proteins identified in both timecourse and replicated analyses concordantly (up/down) and significantly (p<0.05 in the replicated analysis) regulated at 48 hr post-infection. Of course, there is uncertainty associated with any quantitative approach (for example, protein identification false discovery rates) – which does not necessarily detract in practical terms from the usefulness of the dataset.

We would be happy for the manuscript to be reclassified as a research article. As such, we have incorporated the following textual modifications into the revised manuscript:

Abstract

“We used multiplex tandem mass tag (TMT)-based whole cell proteomics to perform a comprehensive time course analysis of >6,500 viral and cellular proteins during HIV infection.”

Introduction

“Here, we provide a comprehensive temporal proteomic analysis of HIV infection.”

Discussion

“In this study, we provide a comprehensive description of temporal changes in >6,500 viral and cellular proteins during HIV infection.”

*2) While the reviewers do appreciate that Figure 4 and Figure 5 firmly establishes PPP2R5 depletion in a 293T co-transfection system requires Cul5, ELOb/C, CBF-β and a functional proteasome, taken as a whole this single model system is not enough to prove that HIV-1 Vif acts specifically on PPP2R5 proteins to induce their degradation in natural infection. Minimally, the authors should provide evidence that HIV-1 Vif does interact with PPP2R5A-E subunits in HIV-infected T cells and forms a complex containing Cul5 components. Furthermore, direct evidence that Vif mediates the degradation of PP2A-B56 subunits in HIV-1 infected T cells should be demonstrated by using pulse-chase experiments or cycloheximide chase assay.*

We considered it unlikely that the mechanism of Vif-dependent depletion of PPP2R5A-E would differ between cell types and systems, but recognise that this was formally possible on the basis of the evidence originally presented. In order to address this possibility and perform the experiments requested, we identified an anti-PPP2R5D antibody (ab88075, Abcam) which works for both intracellular flow cytometry and immunoprecipitation of endogenous PPP2R5D. We are therefore now able to show that:

A) Vif is required for PPP2R5D depletion in HIV-infected primary human CD4^+^ T cells (new Figure 4—figure supplement 2);

B) Vif and PPP2R5D subunits co-immunoprecipitate in 293T cells and HIV-infected CEM-T4 T cells (new Figure 5—figure supplement 1);

C) PPP2R5D is degraded by Vif in HIV-infected CEM-T4 T cells on the basis of both cycloheximide (new Figure 5—figure supplement 2) and [^35^S]methionine/[^35^S]cysteine metabolic labelling/pulse-chase (new Figure 5—figure supplement 3) analyses.

Since it is well established in the literature that Vif forms a complex with Cul5 components in T cells, we have not repeated these experiments in our study (Jager et al., 2012; Yu et al., 2003; Zhang et al., 2012). Nonetheless, to extend the observations from our cycloheximide chase analysis, we confirmed that (as in 293T cells, Figure 5) degradation of PPP2R5D by Vif in HIV-infected CEM-T4 T cells is proteasome-dependent (new Figure 5—figure supplement 2).

In addition to providing the new figure supplements and updating the relevant methods and figure legends, we have described these experiments in the revised manuscript:

Results

“Vif-dependent degradation of PPP2R5 subunits was confirmed by immunoblot of HIV-infected CEM-T4 T cells (PPP2R5A and PPP2R5D; Figure 4,) and intracellular flow cytometry of HIV-infected CEM-T4 and primary human CD4^+^ T cells (PPP2R5D; Figure 4—figure supplement 2).”

“Consistent with a protein-level interaction between Vif and PPP2R5 subunits, we observed co-immunoprecipitation of FLAG-tagged Vif with HA-tagged and endogenous PPP2R5D in 293T cells (Figure 5—figure supplement 1), and co-immunoprecipitation of untagged Vif with endogenous PPP2R5D in CEM-T4 T cells infected with HIV (Figure 5—figure supplement 1). […] Finally, we confirmed using cycloheximide chase (Figure 5—figure supplement 2) and [^35^S]methionine/[^35^S]cysteine metabolic labelling/pulse-chase (Figure 5—figure supplement 3) analyses that degradation of PPP2R5D was accelerated in the presence of Vif in HIV-infected CEM-T4 T cells.”

*3) Figure 7: The requirement for CBF-β function suggests that the canonical APOBEC-degrading Vif ligase is also responsible for degradation of the PPP2R5 proteins. If this is the case, and based on many prior studies showing that Vif is intrinsically disordered, then this interaction should be primate specific and CBF-β requiring. The non-primate lentiviral Vif proteins clearly function without CBF-β so reviewers strongly question the single MVV Vif data point in Figure 7. We strongly suggest removing these data from the paper and temper the claims of complete conservation. Alternatively, repeat the experiment in Figure 7 with SRLV(MVV) plus/minus Vif to prove that the downregulation of PPP2R5 components is indeed Vif-dependent. The claim of conservation is very important and needs to be carefully investigated and justified.*

We agree with the reviewers that this is an important point, and that Vif variants from non-primate lentiviruses (including SRLV) neither interact with CBFβ (Ai et al., 2014; Kane et al., 2015; Yoshikawa et al., 2016; Zhang et al., 2014) nor require CBFβ to antagonise cognate APOBEC proteins (Ai et al., 2014; Kane et al., 2015). Nonetheless, our data clearly demonstrate that SRLV Vif is able to deplete all human PPP2R5 subunits at least as efficiently as NL4-3 Vif (Figure 7).

We therefore hypothesised that, whereas CBFβ is required for PPP2R5 subunit depletion by NL4-3 Vif (Figure 5), PPP2R5 subunit depletion by SRLV would be independent of CBFβ. This is indeed the case (new Figure 7—figure supplement 3).

Furthermore, as with NL4-3 Vif, SRLV Vif co-immunoprecipitates with PPP2R5D (new Figure 7—figure supplement 3).

These data strongly support a model in which primate Vif variants require CBFβ for substrate degradation (PPP2R5A-E and APOBEC proteins), whereas SRLV Vif does not (again, both PPP2R5A-E and APOBEC proteins).

We recognise that the ability of SRLV Vif to deplete PPP2R5 subunits does not guarantee that this ability is also present in Vif variants of all non-primate lentiviruses, and we are therefore careful not to make such a claim. Furthermore, we explicitly state in our discussion that, whilst the simplest hypothesis is that the ability to deplete PPP2R5 subunits was present in the common ancestor of primate and SRLV Vif variants, it is theoretically possible that this ability could have arisen independently (i.e. convergent evolution).

In addition to providing the new figure supplements and updating the relevant methods and figure legends, we have described the new experiments in the revised manuscript:

Results

“Finally, we focused specifically on the distantly related SRLV and NL4-3 (HIV-1) Vif variants. Vif-dependent antagonism of APOBEC proteins shows lineage-specificity, and SRLV Vif is unable to antagonize human APOBEC3G (Larue et al., 2010). […] As with APOBEC proteins, we found CBFβ but not EloB to be dispensable for degradation of HA-PPP2R5E by SRLV Vif (Figure 7—figure supplement 3).”

Furthermore, to ensure there is no ambiguity, we have tempered our abstract and discussion on this point as suggested by the reviewers:

Abstract

"The ability of Vif to target PPP2R5 subunits is found in primate and non-primate lentiviral lineages, and remodeling of the cellular phosphoproteome is therefore a second ancient and conserved Vif function."

Discussion

“Here, we show that degradation of PP2A-B56 subunits is conserved across Vif variants from diverse HIV and SIV lentiviruses of primates, as well as a small ruminant lentivirus of sheep (SRLV). […] Alternatively, Vif variants from these lineages may have independently acquired this ability. Either possibility strongly suggests a critical selective advantage for lentiviral replication or persistence in vivo.”